# How self-governance willingness and participation efficacy shape residents' satisfaction with urban public services: Evidence from neighborhood renewal in Hangzhou, China

Yan Sun[1,2], Wenjie Hu[1,3,2]*, Xinqu Xia[1,2]

**1** School of Law, Hangzhou City University, Hangzhou, Zhejiang, China, **2** Institute of Urban Development and Strategy, Hangzhou City University, Hangzhou, Zhejiang, China, **3** School of Economics, Zhejiang University, Hangzhou, Zhejiang, China

* huwenjie@zju.edu.cn

## Abstract

Using an original survey of 2,202 households across 36 renewed neighborhoods in Hangzhou, China, this study examines how resident agency—operationalized as self-governance willingness and participation efficacy—shapes satisfaction with urban public services within a legally structured participatory governance context. To address endogeneity, we estimate two-stage least squares models with district and street fixed effects, instrumenting self-governance willingness with distance to the provincial government offices and participation efficacy with policy awareness and community involvement; instrument diagnostics indicate strong and valid instruments. In the second stage, both constructs are positively associated with satisfaction. Mediation analyses show that participation efficacy is associated with lower perceived disparities between renewed and newly built neighborhoods, and perceived disparities, in turn, are negatively related to satisfaction, consistent with partial mediation in which effective participatory channels translate engagement into experienced fairness. By contrast, self-governance willingness is positively related to disparity sensitivity, yielding a suppression pattern absent adequate institutional responsiveness. Results are robust to additive-index specifications and the original five-point outcome, and with significantly larger effects among low-income, urban-hukou, small-unit groups. The findings underscore the value of institutionalized, responsive participatory arrangements that align with resident preferences, clarify responsibility allocation, and sustain feedback loops to enhance procedural fairness and, in turn, satisfaction within legally structured neighborhood renewal.

**Data availability statement:** All relevant data are within the manuscript and its Supporting Information files.

**Funding:** This work was supported by the National Social Science Fund of China (Grant No. 20BZZ040); there was no additional external funding for this study. The funder provided financial support only and had no role in study design, data collection, analysis or interpretation of data, decision to publish, or preparation of the manuscript. The authors retained full independence and had full access to all data.

**Competing interests:** The authors have declared that no competing interests exist.

# 1. Introduction

Public services satisfaction is widely recognized as a critical indicator of urban governance effectiveness [1–3], shaping residents' evaluations of service quality, perceptions of government responsiveness, and overall well-being [4, 5]. While existing research has extensively examined satisfaction from a government-centric perspective—emphasizing macro-level determinants such as economic development, fiscal investment, and administrative capacity [6, 7], as well as micro-level attributes such as income, education, and residential status [8, 9]—the role of residents as active participants in governance processes has remained comparatively underexamined. This gap is particularly salient in contexts where community engagement plays a central role in shaping service outcomes [10, 11, 12]. As urban governance frameworks in China undergo increasing institutional transformation [13, 14], residents' self-governance willingness and participation efficacy have emerged as critical—yet empirically underexamined—determinants of public services satisfaction.

Grassroots participatory governance has gained increasing prominence globally as an alternative governance mechanism [15, 16], aligning with broader shifts toward participatory decision-making and deliberative democracy [17, 18]. Theoretically, participatory governance is often examined through the lens of citizenship and institutional participation, highlighting how civic engagement and bottom-up governance structures can foster accountability, responsiveness, and legitimacy in public services delivery [19, 20]. Empirically, in developed democracies such as the United States and the United Kingdom, active citizen participation has demonstrably shaped the trajectory of state reform and improved the quality of democratic life [21]. Evidence from developing countries, while largely positive, remains limited in scope and methodological rigor [22]. In China, while symbolic participation has long been a prominent feature of grassroots governance [23], recent years have witnessed substantive progress, with participatory governance increasingly institutionalized through consultative democracy platforms, participatory budgeting, and localized governance mechanisms [24–26], reflecting a growing emphasis on resident-driven governance as a means to enhance procedural legitimacy in service delivery [27].

Considering these developments, this study posits that two resident-oriented constructs—self-governance willingness and participation efficacy—jointly influence satisfaction with urban public services. Self-governance willingness refers to residents' proactive intent to assume responsibility in key aspects of neighborhood renewal, including decision-making, performance evaluation, and property management selection [28, 29]. Participation efficacy captures residents' perceived effectiveness of their involvement—whether their input is valued, acknowledged, and translated into concrete governance outcomes [30–32]. Theoretically, higher levels of both constructs are expected to strengthen governance legitimacy and perceptions of procedural fairness, build social trust, and enhance institutional responsiveness [33, 34], thereby improving public satisfaction with service delivery [35, 36]. However, in the Chinese context, policy-led efforts to institutionalize participatory governance often confront persistent structural constraints, limited feedback mechanisms, and bureaucratic

inertia [37–39]. These challenges raise critical concerns about whether these resident capacities can be meaningfully actualized in practice [40, 41].

The renewal of old urban neighborhoods provides a compelling empirical context for investigating evolving governance dynamics. Typically constructed prior to 2000 under the state-led work-unit (*danwei*) system, these neighborhoods face persistent challenges such as aging infrastructure, deficient public services, and deteriorating living conditions [42–44]. Since 2019, neighborhood renewal has been elevated to a national policy priority, signaling a broader shift from state-centered urban management toward more inclusive, participatory models of grassroots governance [45]. In particular, the 2020 directive issued by the General Office of the State Council institutionalized participatory planning by mandating public consultations, resident deliberations, and sustained civic engagement throughout the renewal process. This policy evolution marks a significant departure from earlier top-down approaches, emphasizing co-production and democratic responsiveness as key pillars of China's contemporary urban governance reforms.

Hangzhou represents a particularly illustrative case within this national transformation. Launched in 2019, its neighborhood renewal program had completed more than 1,400 projects by the end of 2024—one of the largest such initiatives in China. This exceptional scale, combined with Hangzhou's distinctive governance architecture, makes it an especially relevant case for examining participatory urban governance. The program features substantial fiscal investment, broad community coverage, and high public visibility, reflecting a strong governmental commitment to improving local residential environments. Crucially, Hangzhou has embedded a model of resident voluntarism within its policy framework, whereby decisions regarding whether, what, and how to renovate—as well as the post-renewal governance of neighborhood affairs—are determined through collective resident deliberation. Each project must satisfy the Property Law's double two-thirds rule—approval by at least two-thirds of owners and representing at least two-thirds of total floor area—thereby ensuring procedural legitimacy and substantive public consent. Fieldwork further confirms that residents have been actively involved across all stages of the renewal process—including agenda setting, implementation oversight, and outcome evaluation—facilitating a notable shift from symbolic participation to substantive civic engagement.

Despite the growing scholarly interest in participatory governance and public services satisfaction, three precise research gaps remain. First, micro-level causal evidence on how distinct dimensions of resident agency—self-governance willingness and participation efficacy—shape service satisfaction is scarce; existing work privileges macro policy determinants or treats "participation" as a monolith, leaving the resident-side mechanism underspecified and often coarsely operationalized [46, 47]. Second, the consequences of institutionalized participatory arrangements for residents' perceptions of neighborhood quality and procedural/distributive fairness—particularly within legally structured Chinese urban renewal—are not well established, and the mediating role of perceived neighborhood disparities has not been rigorously identified. Third, the evidence base remains dominated by small-N qualitative cases or normative discussions with limited identification strategies, constraining both causal inference and external validity.

To address these gaps, this study empirically examines the causal effects of self-governance willingness and participation efficacy on residents' satisfaction with urban public services in the context of Hangzhou's neighborhood renewal initiatives. Drawing on survey data collected in September 2024 from 2,202 residents across 36 renewed neighborhoods, the study engages four interrelated research questions. First, how can the causal relationships between self-governance willingness, participation efficacy, and public services satisfaction be rigorously identified? Second, in what ways do self-governance willingness—capturing residents' involvement in decision-making, performance evaluation, and property management—and participation efficacy—reflecting their perceived influence in pre-decision consultation, process supervision, and feedback responsiveness—shape satisfaction with public service outcomes? Third, does perceived disparity between renewed and newly built neighborhoods serve as a mediating mechanism linking participation dimensions to satisfaction? Finally, how do these relationships vary across resident subgroups defined by income, hukou status, housing area, and housing ownership type?

This study advances the literature on participatory governance and public services delivery in three significant respects. Conceptually, it shifts the analytic lens from government-centric accounts to a resident-centered perspective, foregrounding resident agency—operationalized as self-governance willingness and participation efficacy—as a primary mechanism of neighborhood-level outcomes. We articulate a parsimonious causal pathway in which resident agency shapes public services satisfaction in part through perceived neighborhood disparities: participation efficacy tends to mitigate perceived disparities and thereby elevate satisfaction, whereas self-governance willingness can heighten sensitivity to disparities in the absence of institutional responsiveness, suppressing its direct positive association with satisfaction. Methodologically, the study leverages a large resident-level dataset—2,202 valid surveys from 36 renewed neighborhoods in Hangzhou—to deliver micro-foundational evidence with greater external validity than small-N or purely qualitative designs. It constructs multidimensional behavioral indicators spanning the full participatory cycle and measures the breadth and depth of resident agency via factor analysis and complementary index construction. Causal identification relies on a two-stage least squares instrumental-variables strategy; mechanism testing employs mediation analysis centered on perceived neighborhood disparity; and distributional relevance is examined through heterogeneity analyses across income, hukou status, housing area, and ownership type. Robustness checks using alternative operationalizations of key constructs and the original five-point satisfaction outcome corroborate all core results. Practically, the findings show that institutionalized participatory arrangements—when aligned with resident preferences, embedded in clear governance responsibilities, and implemented within a legally structured renewal program—are associated with substantive improvements in perceived procedural fairness and distributive equity, ultimately strengthening public trust and subjective well-being. The evidence clarifies where and how resident agency can be translated into better service evaluations in comparable urban contexts.

The structure of the paper is as follows. Section 2 introduces the analytical framework, formulates the core hypotheses, and outlines the empirical strategy. Section 3 describes the dataset and methodological approach, including survey design, sampling procedures, and variable construction. Section 4 presents the empirical results. Section 5 examines heterogeneity across income, hukou status, housing size, and housing ownership type. Section 6 discusses the theoretical and practical implications of the findings, while addressing key limitations. Section 7 concludes with a summary of key insights and policy implications.

## 2. Theoretical analysis and research hypotheses

### 2.1. Theoretical framework

Grounded in both theoretical literature and empirical practices from Hangzhou's neighborhood renewal initiatives, this study conceptualizes self-governance willingness as residents' readiness to assume governance responsibilities across three key stages of the renewal process: (1) decision-making responsibility—the willingness to participate in the formulation of renovation plans; (2) performance evaluation responsibility—the willingness to assess the effectiveness of implementation; and (3) property management selection responsibility—the willingness to engage in decisions regarding post-renewal property management. These dimensions correspond to sequential governance phases—initiation, oversight, and sustainability—and collectively reflect the depth of residents' engagement. Proactive involvement across these stages is theorized to shape satisfaction by enhancing perceptions of procedural fairness, governance transparency, and service responsiveness.

In parallel, this study defines participation efficacy along three dimensions that reflect residents' subjective assessment of their engagement experience: (1) pre-decision participation, or the degree to which residents feel meaningfully involved in deliberations prior to decision finalization; (2) process supervision, or the perceived ability to monitor and influence project implementation; and (3) feedback responsiveness, or the extent to which authorities acknowledge and act upon residents' input. These constructs mirror Hangzhou's institutional design, which promotes participatory governance throughout planning, execution, and evaluation. However, implementation gaps remain salient:

although policy frameworks formally mandate participation, actual practices often diverge from these ideals. This mismatch highlights the distinction between formal inclusion and substantive participation, suggesting that the quality of engagement—rather than its mere procedural existence—plays a decisive role in shaping residents' satisfaction with urban public services.

Beyond the direct effects of self-governance willingness and participation efficacy, this study introduces perceived neighborhood disparity as a mediating variable, drawing on relative deprivation theory [48, 49]. Relative deprivation theory suggests that individuals assess their well-being comparatively rather than absolute terms, and that perceived inequities in relative standing strongly shape attitudes toward institutions [50, 51]. In the context of urban renewal, residents may view their neighborhoods as disadvantaged when renewed areas lag behind newly developed ones in terms of amenities, green space, or property management quality. Consequently, even when tangible improvements are made, visible disparities between renewed and newly built neighborhoods can generate dissatisfaction, as residents interpret such gaps as signals of distributive unfairness in service provision.

Building on these theoretical insights, this study proposes an integrated analytical framework that connects self-governance willingness, participation efficacy, perceived neighborhood disparities, and satisfaction with urban public services (Fig 1).

## 2.2. Research hypotheses

This study investigates the effects of residents' self-governance willingness—operationalized through their engagement in decision-making, performance evaluation, and property management selection—and participation efficacy—capturing perceived involvement in pre-decision deliberations, process supervision, and feedback responsiveness—on satisfaction with urban public services. Accordingly, the following hypotheses are proposed:

H1: Higher levels of self-governance willingness are positively associated with residents' satisfaction with urban public services.

H2: Higher participation efficacy exerts a positive influence on residents' satisfaction with urban public services.

H3: Perceived disparities between renewed and newly built neighborhoods mediate the relationship between self-governance willingness, participation efficacy, and satisfaction with urban public services.

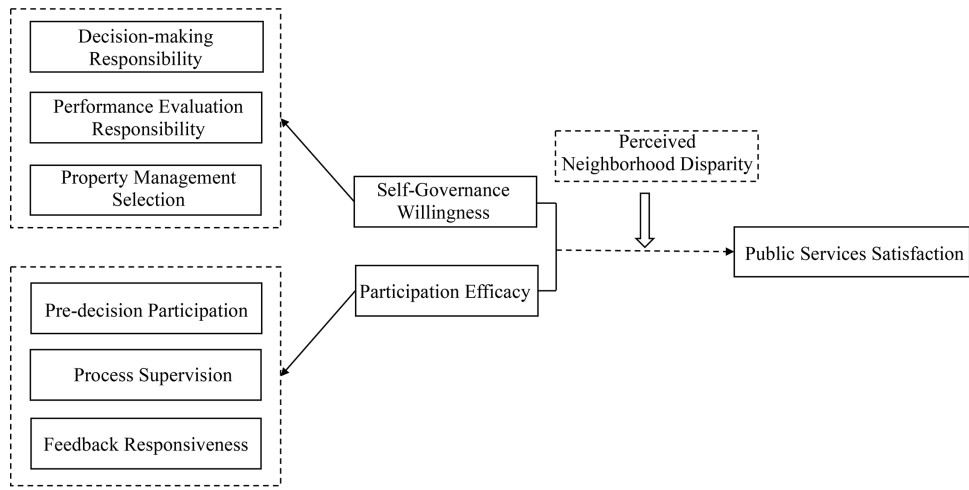

**Fig 1. Analytical framework.**

## 2.3. Empirical model specification

To empirically test the hypothesized relationships, we employ a mediation analysis framework, using perceived neighborhood disparities as the mediating variable. Following the classic three-step approach [52], we estimate the following equations:

$$Satisfaction_{dsp} = \alpha_0 + \alpha_1 Willingness_{dsp} + \alpha_2 Efficacy_{dsp} + \alpha_3 CV_{dsp} + \mu_{1d} + \eta_{1s} + \varepsilon_{dsp} \tag{1}$$

$$Gap_{dsp} = \beta_0 + \beta_1 Willingness_{dsp} + \beta_2 Efficacy_{dsp} + \beta_3 CV_{dsp} + \mu_{2d} + \eta_{2s} + \epsilon_{dsp} \tag{2}$$

$$Satisfaction_{dsp} = \lambda_0 + \lambda_1 Willingness_{dsp} + \lambda_2 Efficacy_{dsp} + \lambda_3 Gap_{dsp} + \beta_4 CV_{dsp} + \mu_{3d} + \eta_{3s} + \zeta_{dsp} \tag{3}$$

Here, the subscript $p$, $d$ and $s$ index individual respondents, administrative districts, and street-level units, respectively. The dependent variable $Satisfaction_{dsp}$ measures residents' satisfaction with urban public services. The key explanatory variables $Willingness_{dsp}$ and $Efficacy_{dsp}$, measure self-governance willingness and participation efficacy, respectively. $CV_{dsp}$ is a vector of control variables. District and street fixed effects are denoted by $\mu_{id}$ and $\eta_{is}$, respectively. Coefficients $\alpha_1$ and $\alpha_2$ represent the total effects of self-governance willingness and participation efficacy on satisfaction, while $\lambda_1$ and $\lambda_2$ indicate their direct effects, net of mediation. The product terms $\beta_1 \times \lambda_3$ and $\beta_2 \times \lambda_3$ capture the indirect (mediated) effects transmitted through residents' perceived disparities between renewed and comparable newly built neighborhoods.

To determine the presence and nature of mediation, we follow the sequential test strategy [52, 53]. First, we examine whether the total effects ($\alpha_1$ and $\alpha_2$) are statistically significant. If neither reaches significance, mediation is ruled out. If at least one total effect is significant, we proceed to test the significance of the indirect path components $\beta_1$ and $\lambda_3$ (or $\beta_2$ and $\lambda_3$). If both are significant, we then test whether the corresponding direct effect ($\lambda_1$ or $\lambda_2$) remains significant. A significant direct effect implies partial mediation, whereas a non-significant direct effect indicates full mediation. If either β or $\lambda_3$ is not significant, we apply the Sobel test to assess the statistical significance of the indirect effect. A significant Sobel result confirms the presence of a mediation effect; otherwise, mediation is not supported.

## 3. Materials and methods

### 3.1. Survey design and sampling

This study draws on original survey data collected in September 2024 from households residing in renewed old residential neighborhoods in Hangzhou, China. According to data from the Hangzhou Municipal Urban–Rural Development and Management Service Center, a total of 233 neighborhoods underwent official renewal in 2023. To ensure the representativeness of the sample, a stratified sampling strategy was adopted, considering neighborhood characteristics such as construction period, average housing unit size, and geographic location. Based on these criteria, 36 neighborhoods—accounting for approximately 16.3% of the citywide renewed communities—were selected for inclusion in the survey.

The sample size for each neighborhood was proportionally allocated to the total number of residential units (≥2,000 = 100 questionnaires; 1,000–1,999 = 80; 500–999 = 60; <500 = 40). Data were collected through face-to-face interviews by 20 trained undergraduate enumerators organized into five subgroups (each covering 7–8 neighborhoods), with one team leader per neighborhood supervising implementation and conducting spot checks. Before any questions were asked, enumerators obtained informed verbal consent by reading verbatim the standardized opening printed at the top of the questionnaire and proceeded only after explicit agreement. Consent was documented on the instrument (checkbox, date/time, enumerator ID) and witnessed by the team leader, who countersigned daily field logs. Only adults (≥18 years) were interviewed; no minors were enrolled. All responses were anonymized at collection and further de-identified prior to analysis.

In total, 2,300 questionnaires were distributed and successfully collected on-site, yielding a response rate of 100%. After excluding incomplete or invalid responses, 2,202 questionnaires were retained for empirical analysis, resulting in an effective response rate of 95.7%. The survey instrument comprised three modules: (1) Demographic and socioeconomic characteristics, including gender, age, educational attainment, occupation, and household income; (2) Evaluations of renewal outcomes, focusing on perceptions of neighborhood environment, infrastructure upgrades, and satisfaction with public services; (3) Governance engagement, with emphasis on residents' self-governance willingness and participation efficacy.

To account for spatial heterogeneity in renewal intensity, the sampling strategy was further refined to reflect district-level variation. Four districts—*Shangcheng, Linping, Gongshu*, and *Xihu*—accounted for 79.4% of all renewed projects completed in 2023 and were therefore prioritized in the sampling design. These districts collectively contributed 75% of the total sampled neighborhoods, ensuring sufficient variation in local governance practices and service satisfaction across different urban contexts. Table 1 summarizes the distribution of sampled neighborhoods, administered questionnaires, and valid responses across the ten districts included in the survey.

## 3.2. Variable selection

The dependent variable, public services satisfaction, captures residents' overall evaluation of renewal outcomes. Respondents initially evaluated their satisfaction with urban public services using a five-point Likert scale ranging from "very dissatisfied" to "very satisfied." For empirical analysis, this variable was recoded into a binary outcome ($0 =$ dissatisfied, $1 =$ satisfied).

The independent variable self-governance willingness is constructed as a latent factor derived from exploratory factor analysis (EFA) of three binary indicators that reflect residents' preferences regarding responsibility allocation across key stages of neighborhood renewal: decision-making, performance evaluation, and property management selection. Specifically, Decision-making responsibility was measured by the question, "Who do you think should be primarily responsible for making major decisions regarding neighborhood renewal??" ($0 =$ government; $1 =$ residents or resident-led organizations. Performance evaluation responsibility was assessed by asking, "Who should be primarily responsible for evaluating the outcomes of the neighborhood renewal project?" ($0 =$ government, $1 =$ residents or resident-led organizations). Property management selection responsibility was measured by asking, "Who should be primarily responsible for selecting the

**Table 1. Summary of sampled neighborhoods and questionnaire distribution.**

|  | District | Renewed neighborhoods (2023) | Total Housing Units | Sampled neighborhoods | Distributed Questionnaires | Proportion (%) |
|---|---|---|---|---|---|---|
| 1 | Shangcheng | 135 | 42606 | 13 | 880 | 36.11% |
| 2 | Linping | 26 | 5061 | 5 | 260 | 13.89% |
| 3 | Xihu | 17 | 6058 | 4 | 280 | 11.11% |
| 4 | Tonglu | 15 | 1095 | 0 | 0 | 0.00% |
| 5 | Yuhang | 10 | 1062 | 2 | 100 | 5.56% |
| 6 | Jiande | 8 | 3225 | 2 | 100 | 5.56% |
| 7 | Gongshu | 7 | 6220 | 5 | 340 | 13.89% |
| 8 | Xiaoshan | 6 | 4082 | 2 | 160 | 5.56% |
| 9 | Qiantang | 3 | 720 | 1 | 40 | 2.78% |
| 10 | Fuyang | 3 | 1905 | 1 | 60 | 2.78% |
| 11 | Binjiang | 2 | 1876 | 1 | 80 | 2.78% |
| 12 | Xihu Scenic Area | 1 | 295 | 0 | 0 | 0.00% |
| 13 | **Total** | **233** | **74205** | **36** | **2300** | **100.00%** |

property management provider after the completion of the renewal project?" (0 = government, 1 = residents or resident-led organizations). Exploratory factor analysis results (Table 2) indicate that the three observed indicators load strongly onto a single latent dimension. The first principal component yields an eigenvalue of 1.585, accounting for approximately 52.8% of the total variance. The Kaiser–Meyer–Olkin (KMO) measure of sampling adequacy is 0.612, and Bartlett's test of sphericity is statistically significant ($\chi^2 = 527.210$, $p < 0.01$), confirming the appropriateness of the data for factor analysis. These findings support the interpretation of the factor as a unidimensional construct representing self-governance willingness.

The independent variable, participation efficacy, is conceptualized as a latent construct capturing residents' perceived influence and engagement in participatory governance throughout the renewal process. It is operationalized using three binary indicators: pre-decision participation, process supervision, and feedback responsiveness. Specifically, pre-decision participation was measured by asking, "Were residents consulted before the renewal plan was finalized?" (0 = no, 1 = yes); process supervision by "Did residents take part in supervising the implementation of the renewal project?" (0 = no, 1 = yes); and feedback responsiveness by "Were residents' suggestions or complaints effectively addressed during the process?" (0 = no, 1 = yes). As reported in Table 2, the three items load strongly onto a single factor, with the first principal component explaining 60.5% of the total variance (eigenvalue = 1.814). The KMO measure of 0.648 and a highly significant Bartlett's test of sphericity ($\chi^2 = 1027.199$, $p < 0.01$) confirm the suitability of the data for factor analysis. These results support the interpretation of participation efficacy as a coherent unidimensional construct reflecting residents' perceived agency in participatory governance.

To evaluate potential common-method variance (CMV) arising from self-reported data, all measurement items across the latent constructs were included in Harman's single-factor test using unrotated exploratory factor analysis. The first factor accounted for 30.4% of the total variance (see detailed results in S1 Table), which is well below the conventional 40% threshold, indicating that CMV is unlikely to bias the results. Together with the construct-level EFA findings, this provides additional evidence supporting the validity and robustness of the measurement model.

The mediating variable, perceived neighborhood disparity, captures residents' subjective assessment of service quality gaps between their renewed neighborhoods and comparable newly developed communities. This perception is measured on a three-point ordinal scale (0 = low, 1 = moderate, 2 = high disparity), reflecting the relative evaluation of neighborhood infrastructure, amenities, and overall service provision.

To reduce potential confounding bias, the model incorporates a comprehensive set of control variables reflecting demographic, socioeconomic, and neighborhood-specific attributes. These include gender, age, education, employment type, household registration status (*hukou*), household size, presence of children or elderly family members, annual household income, housing area, housing type, year of construction, length of residence, perceived neighborhood safety, perceived quality of neighborly relations, presence of community-based organizations, and intention to relocate within five years.

Table 3 presents descriptive statistics for all key variables. On average, satisfaction with public services is moderate (mean = 0.458, SD = 0.498). Levels of self-governance willingness are generally high, particularly regarding performance evaluation responsibility (mean = 0.913, SD = 0.282), suggesting widespread support for community involvement in

**Table 2. Principal component analysis results.**

|  | Factor | Eigenvalue | Difference | Proportion | Cumulative | KMO | Bartlett's test $\chi^2$ |
|---|---|---|---|---|---|---|---|
| Self-governance willingness | Factor 1 | 1.585 | 0.811 | 0.528 | 0.528 | 0.612 | 527.210 |
|  | Factor 2 | 0.773 | 0.131 | 0.258 | 0.786 |  |  |
|  | Factor 3 | 0.642 | . | 0.214 | 1.000 |  |  |
| Participation efficacy | Factor 1 | 1.814 | 1.143 | 0.605 | 0.605 | 0.648 | 1027.199 |
|  | Factor 2 | 0.671 | 0.156 | 0.224 | 0.828 |  |  |
|  | Factor 3 | 0.515 | . | 0.172 | 1.000 |  |  |

**Table 3. Descriptive statistics of key variables.**

| Variables | Definition and Coding | Mean | SD | Min | Max | N |
|---|---|---|---|---|---|---|
| Public services satisfaction | 0 = dissatisfied, 1 = satisfied | 0.458 | 0.498 | 0 | 1 | 2,202 |
| Decision-making responsibility | 0 = government,1 = residents | 0.777 | 0.417 | 0 | 1 | 2,202 |
| Performance evaluation responsibility | 0 = government,1 = residents | 0.913 | 0.282 | 0 | 1 | 2,202 |
| Property management selection responsibility | 0 = government,1 = residents | 0.829 | 0.377 | 0 | 1 | 2,202 |
| Pre-decision participation | 0 = no, 1 = yes | 0.488 | 0.500 | 0 | 1 | 2,202 |
| Process supervision | 0 = no, 1 = yes | 0.272 | 0.445 | 0 | 1 | 2,202 |
| Feedback responsiveness | 0 = no, 1 = yes | 0.352 | 0.478 | 0 | 1 | 2,202 |
| Gender | 0 = female, 1 = male | 0.594 | 0.491 | 0 | 1 | 2,202 |
| Age | 0 = ≤35, 1 = 36-45, 2 = 46-55, 3 = 56-60, 4 = >60 | 2.391 | 1.540 | 0 | 4 | 2,202 |
| Education | 0 = ≤junior high, 1 = high school, 2 = college, 3 = graduate | 1.182 | 0.934 | 0 | 3 | 2,202 |
| Employment sector | 0 = Unemployed or retired,1 = Public sector,2 = Private sector | 0.834 | 0.870 | 0 | 2 | 2,202 |
| Hukou status | 0 = Rural hukou,1 = Urban hukou | 0.824 | 0.381 | 0 | 1 | 2,202 |
| Household size | Number of household members:0 = 1,1 = 2,2 = 3,3 = 4,4 = 5 or more | 1.929 | 1.173 | 0 | 4 | 2,202 |
| Presence of children in household | 0 = no, 1 = yes | 0.434 | 0.496 | 0 | 1 | 2,202 |
| Presence of elderly in household | 0 = no, 1 = yes | 0.708 | 0.455 | 0 | 1 | 2,202 |
| Annual household income | 0 = < 120,000 CNY; 1 = ≥ 120,000 CNY | 0.561 | 0.496 | 0 | 1 | 2,202 |
| Length of residence | 0 = Less than 5 years,1 = 5 years or more | 0.768 | 0.422 | 0 | 1 | 2,202 |
| Housing type | 0 = Affordable resettlement housing,1 = Commercial housing,2 = Reform-era housing | 1.316 | 0.776 | 0 | 2 | 2,202 |
| Year of construction | 0 = Before 1980,1 = 1980–1999,2 = 2000 and after | 0.820 | 0.822 | 0 | 2 | 2,202 |
| Housing area | 0 = ≤90 m², 1 = >90 m² | 0.283 | 0.451 | 0 | 1 | 2,202 |
| Perceived neighborhood disparity | 0 = Low disparity,1 = Moderate disparity,2 = High disparity | 1.260 | 0.858 | 0 | 2 | 2,202 |
| Quality of neighborly relations | 0 = Dissatisfied,1 = Neutral,2 = Satisfied | 1.634 | 0.523 | 0 | 2 | 2,202 |
| Perceived neighborhood safety | 0 = Unsafe,1 = Neutral,2 = Safe | 1.593 | 0.612 | 0 | 2 | 2,202 |
| Presence of community associations | 0 = No, 1 = Yes | 0.331 | 0.471 | 0 | 1 | 2,202 |
| Intention to relocate (within 5 years) | 0 = No, 1 = Yes | 0.165 | 0.371 | 0 | 1 | 2,202 |
| distance to the provincial government's administrative office | kilometers (km) | 14.844 | 21.695 | 1 | 145 | 2,202 |

assessing renewal outcomes. Participation efficacy indicators display notable variation: pre-decision participation received the highest score (mean = 0.488), followed by feedback responsiveness (mean = 0.353), while process supervision scored the lowest (mean = 0.272). This pattern highlights an uneven distribution of participatory engagement across stages of the renewal process.

Perceived neighborhood disparity exhibited considerable variation (mean = 1.260, SD = 0.858), indicating that despite physical improvements, many residents continued to perceive relative disadvantages compared to newly constructed developments. Taken together, the findings suggest high levels of governance willingness, moderate participation efficacy with stage-specific imbalances, and persistent concerns over distributive equity—reinforcing the need for a deeper empirical analysis of how participatory governance influences resident satisfaction in the context of urban renewal.

Perceived neighborhood disparity exhibited moderate-to-high variation (mean = 1.260, SD = 0.858), suggesting that while renewal efforts improved living conditions, residents continued to perceive quality gaps compared to newer residential developments. These findings highlight strong self-governance willingness, moderate-to-high perceived participation efficacy, and persistent concerns over service quality disparities, underscoring the need for a deeper examination of how governance engagement affects public services satisfaction in urban renewal contexts.

## 4. Results

### 4.1. Baseline regression

To address potential endogeneity, this study employs a two-stage least squares (2SLS) estimation strategy, correcting for omitted variable bias and simultaneity concerns in the relationship between self-governance willingness, participation efficacy, and satisfaction with urban public services. Identification relies on two key assumptions: (i) instrument relevance—that the instruments are sufficiently correlated with the endogenous regressors; and (ii) instrument exogeneity—that they are uncorrelated with the structural error term, conditional on observed covariates.

Consistent with established empirical strategies [54, 55], we use the straight-line distance from each neighborhood's centroid to the provincial government's administrative office as an instrument for willingness to engage in self-governance. This distance proxies institutional proximity and the intensity of top-down public-goods provision [56, 57], but is unlikely to directly influence residents' satisfaction with public services. Participation efficacy is instrumented using two theoretically grounded proxies of policy attentiveness and civic engagement [58, 59]: (i) respondents' self-reported awareness of recent municipal policy developments in Hangzhou, and (ii) their participation in community-based collective activities.

Standard diagnostic tests affirm the validity of the instrumental variable approach. The Kleibergen–Paap LM statistics (41.029 and 55.252) reject the null hypothesis of underidentification, indicating that the instruments are sufficiently correlated with the endogenous regressors. The Kleibergen–Paap Wald F-statistics (78.196 for self-governance willingness and 29.013 for participation efficacy) both exceed the conventional threshold of 10, alleviating concerns regarding weak instruments. Furthermore, the Hansen J-statistic for the model of self-governance willingness fails to reject the null hypothesis of overidentifying restrictions, suggesting that the instruments are exogenous and correctly excluded from the second-stage equation.

Table 4 presents the second stage estimates of the 2SLS regression, along with the corresponding first-stage results. After controlling for relevant covariates and fixed effects at both the district and street levels, self-governance willingness remains positively and significantly associated with residents' satisfaction with urban public services ($\beta = 0.192$, $p < 0.05$). This finding provides empirical support for Hypothesis H1, suggesting that residents who express stronger preferences for autonomous governance are more likely to report favorable evaluations of neighborhood renewal outcomes. Similarly, participation efficacy exhibits a positive and statistically significant effect on satisfaction ($\beta = 0.155$, $p < 0.01$), even after adjusting for the same set of controls. This result offers strong support for Hypothesis H2 and highlights the importance of not only enabling participation but ensuring its perceived effectiveness. Together, these findings underscore the critical role of resident agency and engagement in shaping public perceptions of service quality within the context of state-led urban renewal. The first-stage results confirm the strength and relevance of the selected instruments. Specifically, distance to the provincial government's administrative office is significantly associated with self-governance willingness ($\beta = 0.007$, $p < 0.01$), while policy awareness ($\beta = -0.229$, $p < 0.01$) and participation in community activities ($\beta = 0.401$, $p < 0.01$) strongly predict participation efficacy.

To address the concern that distance to the provincial government might proxy unobserved spatial amenities or historical investment patterns affecting satisfaction directly, two alternative city-level instruments were employed. Specifically, the straight-line distance from each neighborhood's centroid to the Hangzhou Municipal Government office and the distance to the city's largest railway transportation hub were used. The IV estimates remain consistent in both magnitude and significance when these alternative instruments are applied, confirming that the main findings are robust to alternative specifications and not sensitive to the choice of instrument (see S2 Table for detailed results).

Collectively, the IV estimates provide robust empirical support for the hypothesized causal relationships, demonstrating that both self-governance willingness and participation efficacy exert significant positive effects on residents' satisfaction with urban public services. These findings underscore the dual imperative of enhancing civic agency and institutionalizing participatory mechanisms. Specifically, they call for governance frameworks that not only accommodate residents'

**Table 4. Instrumental variable regression results.**

| | Public Services Satisfaction | | | |
| --- | --- | --- | --- | --- |
| | (1) | (2) | (3) | (4) |
| Panel A: Second stage | | | | |
| Self-governance willingness | 0.266*** | 0.192** | | |
| | (3.24) | (2.45) | | |
| Participation efficacy | | | 0.227*** | 0.155*** |
| | | | (6.23) | (3.41) |
| Panel B: First stage | | | | |
| Distance | 0.006*** | 0.007*** | | |
| | (8.70) | (8.84) | | |
| Policy | | | −0.313*** | −0.229*** |
| | | | (−5.13) | (−3.50) |
| Participation | | | 0.493*** | 0.401*** |
| | | | (7.91) | (6.36) |
| *Kleibergen-Paap rk LM* | 48.524 | 41.029 | 92.790 | 55.254 |
| *Kleibergen-Paap rk Wald F* | 75.647 | 78.196 | 50.555 | 29.013 |
| *Hansen J statistic P-val* | | | 0.753 | 0.850 |
| Control variables | No | Yes | No | Yes |
| District FE | Yes | Yes | Yes | Yes |
| Street-level FE | Yes | Yes | Yes | Yes |
| N | 2202 | 2202 | 2202 | 2202 |
| *R²* | 0.158 | 0.362 | 0.014 | 0.150 |

Notes: *, **, and *** indicate statistical significance at the 10%, 5%, and 1% levels, respectively. All models include district and street fixed effects. Robust t-statistics (in parentheses) are computed using heteroskedasticity-consistent standard errors. Following standard practice in instrumental variable (IV) regressions, the uncentered $R^2$ is reported.

preferences for autonomy but also ensure meaningful, responsive, and inclusive engagement throughout the service delivery process.

## 4.2. Mediation effect regression

To evaluate whether perceived neighborhood disparities mediate the relationship between self-governance willingness, participation efficacy, and satisfaction with urban public services, a series of multivariate linear regressions were conducted. The results are reported in Tables 5 and 6.

Table 5 presents the OLS estimates examining the associations between the key explanatory variables and the proposed mediator—perceived neighborhood disparities. Columns (1) and (3) show that self-governance willingness is positively and significantly associated with perceived disparities, indicating that merely enhancing residents' willingness for self-governance does not reduce feelings of relative deprivation. Instead, it may heighten sensitivity to quality differentials between renewed and newly developed neighborhoods. In contrast, as shown in columns (2) and (3), participation efficacy exhibits a consistently negative and statistically significant association with perceived disparities, suggesting that only residents who perceive their engagement as meaningful are less likely to report such feelings. These findings highlight the distinct roles of governance preferences and participatory experiences in shaping perceptions of spatial equity. While autonomous aspirations alone may exacerbate perceptions of spatial inequity, it is the perceived effectiveness of participation that mitigates such concerns by reinforcing procedural legitimacy and institutional responsiveness.

**Table 5. Effects of self-governance willingness and participation efficacy on perceived neighborhood disparities.**

|  | Perceived Neighborhood Disparity | | |
|---|---|---|---|
|  | (1) | (2) | (3) |
| Self-governance willingness | 0.048*** |  | 0.044*** |
|  | (3.29) |  | (2.92) |
| Participation efficacy |  | −0.062*** | −0.059*** |
|  |  | (−4.27) | (−4.09) |
| Constant | 1.132*** | 1.120*** | 1.122*** |
|  | (5.03) | (4.75) | (4.76) |
| Control variables | Yes | Yes | Yes |
| District FE | Yes | Yes | Yes |
| Street-level FE | Yes | Yes | Yes |
| N | 2202 | 2202 | 2202 |
| $R^2$ | 0.103 | 0.106 | 0.110 |

Notes: *, **, and *** indicate statistical significance at the 10%, 5%, and 1% levels, respectively.

**Table 6. Perceived neighborhood disparities mediating self-governance willingness, participation efficacy, and public services satisfaction.**

|  | Public Services Satisfaction | | | |
|---|---|---|---|---|
|  | (1) | (2) | (3) | (4) |
| Panel A: Second stage |  |  |  |  |
| Self-governance willingness | 0.300*** | 0.249*** |  |  |
|  | (3.63) | (3.07) |  |  |
| Participation efficacy |  |  | 0.197*** | 0.137*** |
|  |  |  | (5.17) | (2.96) |
| Perceived neighborhood disparity | −0.185*** | −0.151*** | −0.111*** | −0.099*** |
|  | (−10.71) | (−8.74) | (−6.93) | (−6.79) |
| Control variables | No | Yes | No | Yes |
| District FE | Yes | Yes | Yes | Yes |
| Street-level FE | Yes | Yes | Yes | Yes |
| N | 2202 | 2202 | 2202 | 2202 |
| $R^2$ | 0.140 | 0.301 | 0.092 | 0.191 |

Notes: *, **, and *** denote significance at the 10%, 5%, and 1% levels, respectively. Sobel test results indicate that the indirect effects corresponding to each variable are statistically significant at the 5% level, supporting the existence of mediating pathways.

Building upon the preceding analyses, this section evaluates whether perceived neighborhood disparities mediate the effects of self-governance willingness and participation efficacy on residents' satisfaction with urban public services. Controlling for a full set of demographic and neighborhood-level covariates, the results remain robust (Table 6). Both self-governance willingness and participation efficacy retain statistically significant positive associations with satisfaction, while perceived disparities exhibit a significant negative effect, offering empirical support for Hypothesis H3.

Interestingly, after adding control variables and fixed effects, the estimated coefficient of participation efficacy declines from 0.155 to 0.137 when perceived neighborhood disparities are introduced, indicating a conventional partial mediation

effect. This suggests that lower perceived disparities enhance the positive impact of participatory engagement on public services satisfaction. In contrast, the coefficient of self-governance willingness increases from 0.192 to 0.249, reflecting a suppressor effect. This counterintuitive pattern arises because self-governance willingness positively predicts perceived neighborhood disparities ($\beta = 0.044$, $p < 0.01$), which in turn negatively affect satisfaction ($\beta = -0.099$, $p < 0.01$). The indirect pathway thereby suppresses the direct positive effect of governance aspirations by channeling dissatisfaction through heightened perceptions of spatial inequity. Once this suppressing mechanism is accounted for, the positive association between self-governance willingness and satisfaction becomes more pronounced. These findings contribute to the literature by revealing that governance preferences and participatory experiences operate through structurally distinct pathways—one potentially heightening dissatisfaction in the absence of institutional responsiveness, the other alleviating perceived injustice through meaningful engagement. This analytical distinction underscores the importance of disentangling residents' governance aspirations from the quality of their participatory experiences when assessing the effectiveness of participatory urban governance.

Notably, respondents who report higher levels of perceived disparities between their renewed neighborhoods and newly developed counterparts express significantly lower satisfaction with public services. This finding reinforces the view that, even under participatory governance frameworks, residents' subjective perceptions of spatial equity remain a key determinant of how they evaluate service outcomes. The mediating role of perceived disparities further highlights that the effectiveness of participatory governance hinges not only on the formal availability of participation channels, but also on how residents perceive and internalize fairness in their everyday living environments.

Taken together, the findings underscore the dual challenge of fostering strong governance aspirations while ensuring that participatory processes are both meaningful and responsive. While participation efficacy improves satisfaction by reducing perceptions of spatial inequality, the positive impact of self-governance willingness may be diminished if not supported by institutional mechanisms that address residents' perceived injustices. These results suggest that narrowing the gap between governance expectations and lived experiences of fairness is essential to realizing the full potential of participatory urban renewal. Institutional strategies to empower residents must also address perceived spatial disadvantages to ensure more equitable urban service delivery.

## 4.3. Robustness tests

To validate the robustness of the baseline findings, this section conducts additional analyses by revising the measurement strategies for both explanatory and outcome variables. Specifically, the latent constructs for self-governance willingness and participation efficacy—originally derived through factor analysis—are replaced with additive indices constructed from their respective binary indicators. Moreover, to address concerns associated with the dichotomization of the dependent variable, the original five-point Likert scale of public services satisfaction is retained in alternative model specifications.

Table 7 presents the results across columns (1) to (4). Regardless of specification, both self-governance willingness and participation efficacy remain positively and significantly associated with satisfaction at the 1% level. These associations hold after controlling for a full set of covariates and incorporating fixed effects at both the district and street levels. The results indicate that the substantive conclusions are not sensitive to alternative operationalizations of the key constructs. Specifically, the coefficients for self-governance willingness range from 0.754 to 0.980, while those for participation efficacy range from 0.362 to 0.501, all statistically significant at conventional levels. Additionally, perceived neighborhood disparity maintains a robust negative association with satisfaction across all specifications, further corroborating its mediating role.

Collectively, these findings affirm the stability and internal validity of the estimated relationships. The positive effects of self-governance preferences and participation efficacy on public services satisfaction are not model-dependent but reflect consistent and empirically grounded patterns of citizen perception and engagement in neighborhood renewal.

**Table 7. Robustness tests results.**

| | Public Services Satisfaction | | | |
|---|---|---|---|---|
| | (1) | (2) | (3) | (4) |
| Panel A: Second stage | | | | |
| Self-governance willingness | 0.980*** | 0.754*** | | |
| | (4.92) | (3.99) | | |
| Participation efficacy | | | 0.501*** | 0.362*** |
| | | | (5.90) | (3.52) |
| Perceived neighborhood disparity | −0.382*** | −0.307*** | −0.230*** | −0.202*** |
| | (−11.86) | (−9.95) | (−7.95) | (−7.57) |
| Control variables | No | Yes | No | Yes |
| District FE | Yes | Yes | Yes | Yes |
| Street-level FE | Yes | Yes | Yes | Yes |
| N | 2202 | 2202 | 2202 | 2202 |
| $R^2$ | 0.773 | 0.828 | 0.103 | 0.215 |

Notes: *, **, and *** denote significance at the 10%, 5%, and 1% levels, respectively.

## 5. Further analysis

### 5.1. Heterogeneity by household income

To examine whether the effects of self-governance willingness and participation efficacy vary across income groups, respondents were categorized into low-income (annual household income below 120,000 CNY) and middle- to high-income groups (120,000 CNY or above). Table 8 reports the regression results for these subsamples.

The results reveal clear income-based heterogeneity in the effects of self-governance willingness. Among low-income households, self-governance willingness is positively and significantly associated with public services satisfaction, indicating that these residents are more responsive to bottom-up governance structures. In contrast, the effect is not statistically significant among higher-income households, suggesting that governance preferences play a less decisive role in shaping satisfaction within this group. This divergence implies that low-income residents may be more sensitive to perceived

**Table 8. Heterogeneity analysis by household income.**

| | < 120,000 CNY | ≥120,000 CNY | < 120,000 CNY | ≥120,000 CNY |
|---|---|---|---|---|
| | (1) | (2) | (3) | (4) |
| Self-governance willingness | 0.225*** | 0.403 | | |
| | (2.62) | (1.35) | | |
| Participation efficacy | | | 0.154** | 0.097 |
| | | | (2.46) | (1.35) |
| Perceived neighborhood disparity | −0.186*** | −0.132*** | −0.104*** | −0.102*** |
| | (−6.09) | (−5.13) | (−4.50) | (−5.31) |
| Control variables | Yes | Yes | Yes | Yes |
| District FE | Yes | Yes | Yes | Yes |
| Street-level FE | Yes | Yes | Yes | Yes |
| N | 966 | 1236 | 966 | 1236 |
| $R^2$ | 0.283 | 0.044 | 0.190 | 0.191 |

Notes: *, **, and *** indicate statistical significance at the 10%, 5%, and 1% levels, respectively.

governance autonomy, potentially due to their greater dependence on public services and fewer residential alternatives. Regarding participation efficacy, the results show a significant positive effect among low-income households, while the coefficient for higher-income households is positive but statistically insignificant. These findings underscore that meaningful resident engagement has a stronger influence on satisfaction among low-income groups, where participatory mechanisms may compensate for limited access to private or market-based services.

Collectively, these findings indicate that both self-governance willingness and participation efficacy are more pronounced among low-income residents. This underscores the need to design participatory governance strategies that are responsive to the priorities and constraints of low-income groups, thereby promoting more equitable, inclusive, and effective outcomes in neighborhood renewal initiatives.

## 5.2. Heterogeneity by household registration (hukou) status

Under China's urban-rural dualistic system, the household registration (hukou) system not only delineates residency status but also structures access to public resources and entitlements [60, 61]. Considering the structural inequalities embedded within the hukou regime, this section evaluates whether the effects of self-governance willingness and participation efficacy differ between urban and rural hukou holders (see Table 9).

The findings reveal substantial hukou-based heterogeneity. Among urban hukou residents, self-governance willingness is positively and significantly associated with satisfaction ($\beta = 0.296$, $p < 0.01$), indicating that those embedded in the formal urban governance apparatus are more responsive to participatory opportunities and prefer autonomy in local decision-making. This likely reflects both higher institutional literacy and greater expectations for procedural inclusion. By contrast, the effect is statistically insignificant among rural hukou residents, suggesting a weaker preference for resident-led governance.

Similar patterns emerge for participation efficacy. Among urban hukou holders, participation efficacy significantly enhances public services satisfaction ($\beta = 0.120$, $p < 0.05$), underscoring the value of institutionalized civic engagement in urban contexts. However, for rural hukou residents, the coefficient is statistically insignificant. This discrepancy may stem from longstanding institutional exclusion and spatial detachment, many rural hukou residents live in urbanized neighborhoods but retain rural registration, limiting their incentives or capacity to engage meaningfully in participatory processes.

**Table 9. Heterogeneity analysis by hukou status.**

|  | Rural | Urban | Rural | Urban |
|---|---|---|---|---|
|  | (1) | (2) | (3) | (4) |
| Self-governance willingness | 0.042 | 0.296*** |  |  |
|  | (0.26) | (3.05) |  |  |
| Participation efficacy |  |  | 0.302 | 0.120** |
|  |  |  | (1.49) | (2.49) |
| Perceived neighborhood disparity | −0.072** | −0.165*** | −0.050 | −0.108*** |
|  | (−2.19) | (−7.96) | (−1.36) | (−6.60) |
| Control variables | Yes | Yes | Yes | Yes |
| District FE | Yes | Yes | Yes | Yes |
| Street-level FE | Yes | Yes | Yes | Yes |
| N | 387 | 1815 | 387 | 1815 |
| $R^2$ | 0.191 | 0.191 | 0.400 | 0.208 |

Notes: *, **, and *** indicate statistical significance at the 10%, 5%, and 1% levels, respectively.

Taken together, these results highlight that participatory governance strategies must be sensitive to hukou-based structural inequalities. While institutionalizing civic engagement can be effective in urban contexts, tailored approaches are needed to activate meaningful participation among rural-registered residents, whose pathways to governance inclusion remain constrained.

## 5.3. Heterogeneity by housing area

Extant research highlights the critical role of housing conditions in shaping residents' governance preferences and participatory behavior [62, 63]. To explore whether household area conditions the effects of self-governance willingness and participation efficacy on public services satisfaction, this section disaggregates the sample into two groups: small-area households (≤ 90 m²) and large-area households (> 90 m²). Table 10 presents the results of subgroup regressions.

Substantial heterogeneity emerges in the relationship between self-governance willingness and satisfaction across housing area. Among residents in smaller units, self-governance willingness is positively and significantly associated with public services satisfaction. In contrast, the coefficient for larger-unit households, although numerically higher, fails to reach statistical significance. This divergence suggests that residents in smaller dwellings—often older and lacking prospects for further improvement—may place greater value on participatory mechanisms to protect and improve their current living conditions. The heightened salience of governance autonomy for this group may stem from their stronger dependence on public investments and limited housing mobility.

In terms of participation efficacy, the results indicate consistently positive effects across both subgroups. Among small-unit households, participation efficacy is significantly associated with satisfaction ($\beta = 0.102$, $p < 0.05$); among larger-unit households, the effect is even stronger ($\beta = 0.280$, $p < 0.10$), albeit at a lower level of statistical significance. These results affirm that the benefits of participatory governance—in terms of enhancing satisfaction—are not confined to disadvantaged housing groups. Rather, civic engagement remains a broadly applicable mechanism for improving perceptions of public services quality across diverse residential contexts.

Together, these findings underscore the differentiated roles of governance autonomy and participatory efficacy in shaping public satisfaction, contingent on housing area. They also reinforce the need for context-sensitive participatory frameworks that accommodate the spatial and structural diversity of urban neighborhoods.

**Table 10. Heterogeneity analysis by housing area.**

| | ≤ 90 m² | >90 m² | ≤ 90 m² | >90 m² |
|---|---|---|---|---|
| | (1) | (2) | (3) | (4) |
| Self-governance willingness | 0.162** | 3.495 | | |
| | (2.06) | (0.52) | | |
| Participation efficacy | | | 0.102** | 0.280* |
| | | | (2.07) | (1.95) |
| Perceived neighborhood disparity | −0.158*** | −0.435 | −0.117*** | −0.070** |
| | (−7.90) | (−0.69) | (−6.82) | (−2.56) |
| Control variables | Yes | Yes | Yes | Yes |
| District FE | Yes | Yes | Yes | Yes |
| Street-level FE | Yes | Yes | Yes | Yes |
| N | 1579 | 623 | 1579 | 623 |
| $R^2$ | 0.411 | 0.219 | 0.194 | 0.027 |

Notes: *, **, and *** indicate statistical significance at the 10%, 5%, and 1% levels, respectively.

## 5.4. Heterogeneity by housing ownership type

As China's urbanization deepens, housing ownership type has become a critical factor shaping community governance preferences and public services delivery outcomes [64, 65]. To explore this relationship, this section examines how variation in housing ownership type—specifically, among residents of affordable resettlement housing, commercial housing, and reform-era housing (*fanggaifang*)—conditions the association between self-governance willingness, participation efficacy, and satisfaction with urban public services (see Table 11).

The findings reveal notable ownership-based heterogeneity. Among residents of reform-era housing, self-governance willingness is positively and significantly associated with satisfaction (β = 0.192, p < 0.05), indicating a heightened preference for autonomous governance structures. This pattern reflects the historical and institutional legacy of these communities: reform-era housing, originally allocated as welfare units under pre-1998 state-owned enterprise (SOE) reforms, often retains high concentrations of former SOE employees. These residents tend to exhibit a stronger sense of collective ownership, identity, and engagement. Furthermore, their low-cost acquisition of property has heightened sensitivity to governance quality and potential value appreciation, making them especially attuned to governance arrangements during neighborhood renewal [66]. In contrast, the effect of self-governance willingness is statistically insignificant among residents of resettlement and commercial housing. Among the former, this may stem from greater reliance on top-down governance due to weaker social capital or institutional embeddedness. Among the latter, governance preferences are often tempered by private expectations of service delivery embedded in market-based housing transactions.

Regarding participation efficacy, the results show a consistently positive and statistically significant effect among commercial housing residents, suggesting that effective civic engagement substantially shapes satisfaction in this group. This reflects a context where residents expect responsiveness and transparency in decision-making processes. While participation efficacy is positively signed among reform-era and resettlement housing residents, the effects are statistically weaker or nonsignificant, potentially due to constrained participation channels or lower expectations of institutional responsiveness.

Taken together, these findings underscore the importance of tailoring participatory governance frameworks to the institutional and historical configurations of different housing communities. Governance reforms must account for residents' embedded experiences and expectations across housing types to enhance the responsiveness, legitimacy, and distributive fairness of urban renewal processes.

**Table 11. Heterogeneity analysis by housing ownership type.**

| | Resettlement | Commercial | Reform-era | Resettlement | Commercial | Reform-era |
| --- | --- | --- | --- | --- | --- | --- |
| | (1) | (2) | (3) | (4) | (5) | (6) |
| Self-governance willingness | −0.024 | 2.989 | 0.192** | | | |
| | (−0.14) | (0.70) | (2.07) | | | |
| Participation efficacy | | | | 0.047 | 0.191*** | 0.090 |
| | | | | (0.41) | (2.99) | (1.13) |
| Perceived neighborhood disparity | −0.148*** | −0.044 | −0.174*** | −0.140*** | −0.068*** | −0.115*** |
| | (−5.40) | (−0.27) | (−6.32) | (−4.12) | (−3.08) | (−4.54) |
| Control variables | Yes | Yes | Yes | Yes | Yes | Yes |
| District FE | Yes | Yes | Yes | Yes | Yes | Yes |
| Street-level FE | Yes | Yes | Yes | Yes | Yes | Yes |
| N | 425 | 657 | 1120 | 425 | 657 | 1120 |
| $R^2$ | 0.230 | −20.461 | 0.384 | 0.256 | 0.149 | 0.176 |

Notes: *, **, and *** indicate statistical significance at the 10%, 5%, and 1% levels, respectively.

### 5.5. Graphical summary of subgroup heterogeneity

To complement the subgroup regressions, Fig 2 presents a compact forest-plot visualization of the IV second stage point estimates with heteroskedasticity-robust 95% confidence intervals across heterogeneity groups. Markers indicate coefficients, whiskers the 95% intervals, and a vertical zero-line aids interpretation. The figure replicates the subgroup patterns reported in Tables 8–11, where intervals for low-income and urban-hukou residents exclude zero for both constructs. By housing area, willingness is significant only for smaller units, whereas efficacy is positive in both and larger in magnitude for larger units. By ownership, willingness is significant in reform-era housing, and efficacy is strongest in commercial housing. All estimates condition on the full covariate set with district and street fixed effects, and subgroup sample sizes match those in the corresponding tables.

Notes: In the > 90 m² and commercial-ownership subgroup, the IV estimate for self-governance willingness is not statistically significant and is imprecisely estimated—yielding an outlying point estimate with a very wide confidence interval; to prevent overinterpretation, this estimate is omitted.

## 6. Discussion

### 6.1. Interpretation of findings

This study addresses several gaps in the participatory governance literature. By embedding resident agency—operationalized as self-governance willingness and participation efficacy—within the specific setting of old-neighborhood renewal, it captures China's broader shift from extensive, greenfield expansion to stock-oriented, quality-enhancing renewal consistent with a people-centered urban development paradigm. Leveraging 2,202 household interviews across 36 renewed neighborhoods in Hangzhou, the study assembles one of the largest third-party datasets of its kind in China and counters long-standing claims that public willingness is weak and participation merely procedural. When core residential interests are implicated, participatory governance emerges as a effective lever for improving frontline public-service performance.

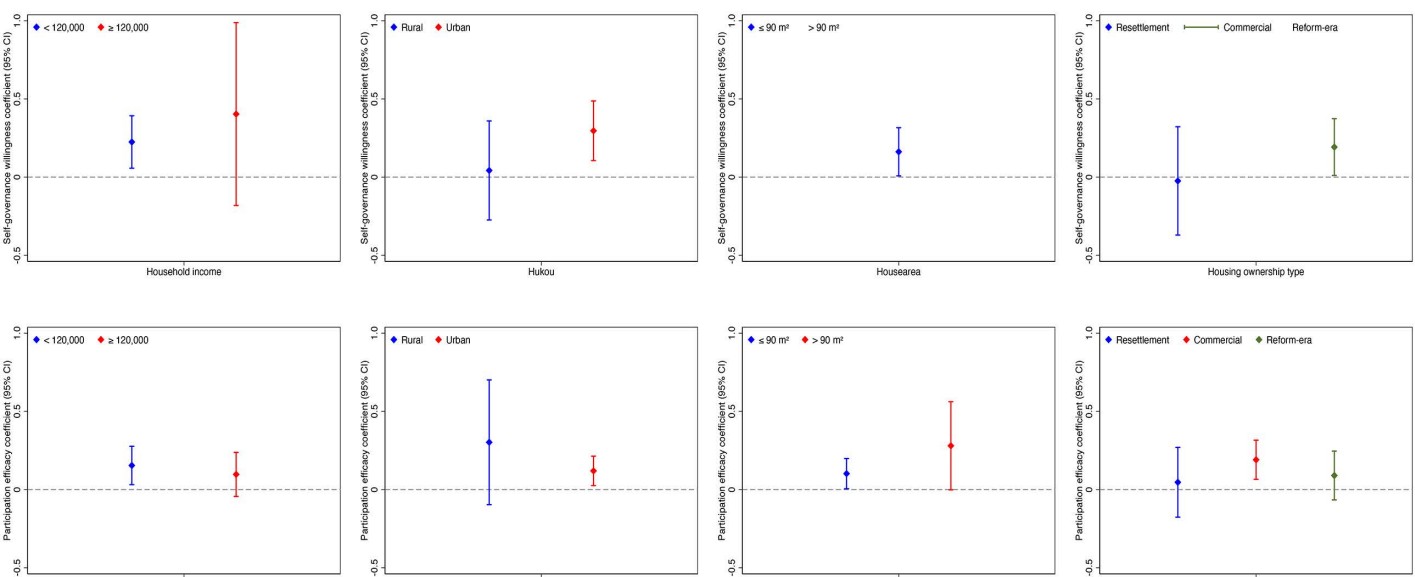

**Fig 2. IV estimates (coefficients and 95% CIs) across heterogeneity subgroups (Top: self-governance willingness; Bottom: participation efficacy).**

Comparative evidence from the United States, the United Kingdom, and Japan underscores the central role of residents in participatory governance, with engagement spanning agenda setting, plan approval, and implementation oversight, thereby helping to mitigate complex social conflicts. In China, the shift toward incremental neighborhood renewal paired with refined governance has embedded resident participation and service provision alongside spatial upgrading. Yet in some locales, a government-led, efficiency-first orientation persists, marginalizing social needs, formalizing participation, and creating an institutional trap of low-cost decision-making followed by high-cost implementation and remediation, with correspondingly low levels of resident satisfaction.

Hangzhou provides a compelling empirical setting. After renewal was elevated to a national priority in 2019, the city launched a comprehensive citywide initiative, and by the end of 2024 it had recorded 2,192 old neighborhoods (≈53.48 million m²), completed renewal in more than 1,400 of them, and produced clear before–after contrasts (Fig 3). At the national level, the 2020 State Council Guiding Opinions call for voluntarism and broad participation. Although the shift from "demolish-and-rebuild" to stock enhancement is evident, multi-actor collaboration often remains nascent, marked by one-way governmental dominance and limited coordination.

Hangzhou addresses these bottlenecks through institutional and procedural innovations. Residents collectively determine whether projects proceed, what they cover, how they are implemented, and how post-renewal affairs are managed, with decisions complying with the resident-consent thresholds mandated by the Property Law. Fieldwork documents engagement across the full project cycle—from agenda setting and plan formulation to implementation oversight and post-renewal stewardship. A multi-actor governance arrangement linking government agencies, subdistrict and community

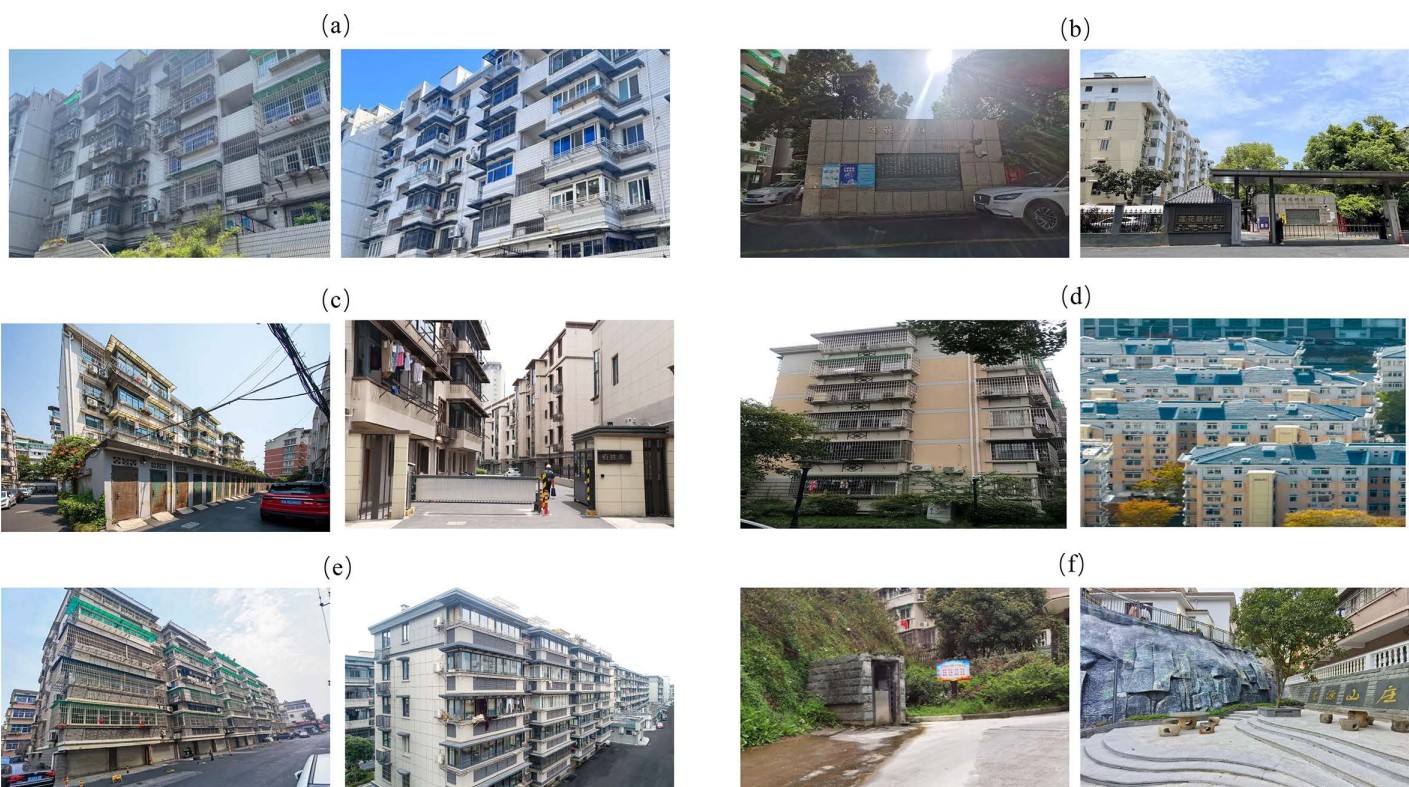

**Fig 3. Representative neighborhoods: before-and-after comparison (Left: pre-renewal; Right: post-renewal). (a)** Gongshu district — Genyuan neighborhood. **(b)** Xihu district — Lianhua Xincun neighborhood. **(c)** Linping district — Baixing Nong neighborhood. **(d)** Binjiang district — Yingchun neighborhood. **(e)** Yuhang district — Taoyuan neighborhood. **(f)** Jiande city — Xinlin neighborhood.

offices, homeowners' organizations, and professional entities in design, construction, property management, and civil-society organizations operates through deliberation, co-design, procedural transparency, and performance evaluation, with collaboration extending from construction into ongoing operations. This arrangement mitigates low participation and weak coordination and provides a scalable template for stock-oriented neighborhood renewal.

The evidence indicates that resident agency materially shapes public services satisfaction in this legally structured renewal setting. Two-stage least squares (2SLS) estimates with district and street fixed effects yield positive second-stage coefficients for self-governance willingness and participation efficacy ($\beta = 0.192$, $p < 0.05$; $\beta = 0.155$, $p < 0.01$; Table 4). Instruments are strong and valid (Kleibergen–Paap rk Wald F = 78.196 for willingness; 29.013 for efficacy; rk LM = 41.029 and 55.254 reject underidentification), and the overidentifying restrictions are not rejected (Hansen J p = 0.753, 0.850). Interpreted as contemporaneous local average treatment effects conditional on the covariates, these estimates shift the account from state-centric technocratic provision to resident-centered governance processes, consistent with participatory governance and procedural justice perspectives in which validation of voice and fair process elevate evaluations of institutional performance.

Mechanism evidence indicates asymmetric roles for the two dimensions of resident agency. Participation efficacy is negatively associated with perceived neighborhood disparities ($\beta = -0.062$ to $-0.059$, $p < 0.01$; Table 5), and disparities, in turn, lower satisfaction ($\beta = -0.185$ to $-0.099$, $p < 0.01$; Table 6). Including disparities reduces the efficacy coefficient from 0.155 to 0.137 ($p < 0.01$), consistent with partial mediation in which effective channels translate engagement into experienced fairness and higher satisfaction. By contrast, self-governance willingness is positively associated with perceived disparities ($\beta = 0.048$ to 0.044, $p < 0.01$; Table 5), yielding a suppression pattern whereby its second-stage coefficient rises from 0.192 to 0.249 ($p < 0.01$) when disparities are controlled (Table 6). This accords with relative-deprivation dynamics—benchmarking renewed neighborhoods against newly built areas can dampen the subjective returns to autonomy preferences unless institutions are sufficiently responsive.

Heterogeneity analyses reveal systematic gradients. Among low-income households, both willingness ($\beta = 0.225$, $p < 0.01$) and efficacy ($\beta = 0.154$, $p < 0.05$) are positive; for higher-income households the corresponding estimates are not significant ($\beta = 0.403$, $p \geq 0.10$; $\beta = 0.097$, $p \geq 0.10$; Table 8). By hukou, effects are evident for urban residents—willingness ($\beta = 0.296$, $p < 0.01$) and efficacy ($\beta = 0.120$, $p < 0.05$)—but not for rural residents ($\beta = 0.042$, $p \geq 0.10$; $\beta = 0.302$, $p \geq 0.10$; Table 9). By housing area, willingness matters in smaller units ($\beta = 0.162$, $p < 0.05$) but not in larger ones; efficacy is positive in both, with a larger estimate in larger units ($\beta = 0.102$, $p < 0.05$ vs. $\beta = 0.280$, $p < 0.10$; Table 10). By ownership, willingness is significant in reform-era housing ($\beta = 0.192$, $p < 0.05$), while efficacy is strongest in commercial housing ($\beta = 0.191$, $p < 0.01$; Table 11). Taken together, these patterns imply greater marginal gains where reliance on public provision is higher, institutional literacy stronger, or collective identities more pronounced. The subgroup estimates are supported by substantial sample sizes (minimum N = 387) and reported standard errors and p-values.

Scope conditions matter. Hangzhou combines substantial fiscal commitment with procedurally binding consent thresholds that tie project initiation and plan approval to resident agreement, thereby reducing the risk of symbolic participation. As a most likely case, it illustrates how resident agency can be activated at scale. Robustness checks show the findings are not artifacts of operationalization. Reconstructing agency as additive indices and estimating models with the original five-point satisfaction outcome yield substantively unchanged results. Perceived disparities remain consistently negative ($-0.382$ to $-0.202$, $p < 0.01$; Table 7). Both self-governance willingness and participation efficacy remain positive and statistically significant at the 1% level (Table 7). Taken together, the results identify a coherent pathway in which participation efficacy reduces perceived disparities and, in turn, raises satisfaction.

## 6.2. Limitations

Despite its contributions, this study has several limitations that point to important directions for future research.

First, the cross-sectional nature of the data, which are based on a one-time survey conducted in 2024, limits our ability to capture temporal dynamics or identify within-household changes over time. Although endogeneity is addressed through an instrumental variable approach, the estimated effects should be interpreted as contemporaneous local average treatment effects (LATEs). The observed impacts may also reflect short-term "halo" responses immediately following the completion of renewal projects, which could weaken over time. Future research should therefore adopt longitudinal panel or pre/post cohort designs to examine temporal dynamics, verify the persistence of these effects, and better address potential reverse causality.

Second, our measures of self-governance willingness and participation efficacy rely on self-reported survey responses, which may be subject to social desirability and recall bias. While the survey timing helped minimize such risks, future studies should incorporate more objective indicators—such as meeting attendance records or actual proposal submissions—and consider mixed-methods approaches to enhance measurement validity.

Third, the study focuses exclusively on Hangzhou, a leading city in participatory governance, which may limit external validity. Comparative research across cities with varying institutional contexts, administrative hierarchies, and participatory structures would help assess the generalizability of the findings and uncover context-specific dynamics.

Fourth, while the study highlights subjective perceptions of fairness, it does not fully explore how spatial inequality and neighborhood heterogeneity shape satisfaction outcomes. Future work should investigate how factors such as income segregation, residential mobility, and spatial fragmentation mediate or moderate the effects of participatory engagement—especially in socially diverse or stratified urban settings. To strengthen causal claims, we also encourage the use of experimental and quasi-experimental designs, such as randomized interventions or difference-in-differences analyses leveraging policy variation.

## 7. Conclusions

Drawing on resident-level data from 2,202 residents in Hangzhou's renewed old neighborhoods, this study investigates how self-governance willingness and participation efficacy shape satisfaction with urban public services. The findings yield four key conclusions:

First, residents' preferences for local governance autonomy are both substantive and multidimensional, spanning decision-making, performance evaluation, and post-renewal management. Higher levels of self-governance willingness are significantly associated with increased satisfaction, underscoring the importance of institutional designs that align with autonomy preferences and enable meaningful bottom-up co-governance in neighborhood settings.

Second, participation efficacy is a robust and consistent driver of satisfaction. When residents perceive their engagement as meaningful and consequential, their evaluations of public services delivery improve markedly. This underscores that the legitimacy of participatory governance depends not only on procedural inclusion but also on the extent to which institutional arrangements confer substantive influence.

Third, the effects of governance engagement operate in part through perceived neighborhood disparities, which attenuate the positive impact of participation efficacy and suppress the influence of self-governance willingness. Residents who perceive their neighborhoods as disadvantaged relative to newly built areas consistently report lower satisfaction, highlighting that formal inclusion must be complemented by visible distributive equity to generate meaningful improvements in evaluations of public-service outcomes.

Fourth, governance preferences and participation effects vary systematically by resident characteristics. Low-income households, urban hukou residents, residents of reform-era housing, and those living in older or smaller units exhibit stronger self-governance preferences and greater responsiveness to participatory engagement. This underscores the need for tailored policy designs calibrated to institutional legacies, resource constraints, and heterogeneous participatory capacities across urban constituencies.

Policy implications follow directly from these conclusions. Neighborhood renewal should move beyond administrative efficiency to institutionalize participatory frameworks that accommodate residents' governance aspirations and localized

autonomy; participatory mechanisms should ensure not only access but also procedural responsiveness, transparency, and continuity across the renewal cycle; and equity considerations should be explicitly integrated—by addressing residents' subjective perceptions of spatial inequality—to enhance perceived legitimacy, reduce dissatisfaction, and promote resident well-being.

## Supporting information

**S1 Data. Raw dataset.**
(DTA)

**S1 Table. Harman's single-factor test.**
(PDF)

**S2 Table. Robustness checks using alternative city-level instruments.**
(PDF)

## Acknowledgments

We thank the undergraduate students at the School of Law, Hangzhou City University, for assistance with questionnaire distribution and data collection. We are also grateful to the Hangzhou Urban–Rural Development and Management Service Center for access to non-confidential policy documents and technical guidelines on old-neighborhood renewal and for facilitating site access and interviews.

## Author contributions

**Conceptualization:** Yan Sun, Wenjie Hu.

**Data curation:** Wenjie Hu.

**Formal analysis:** Wenjie Hu.

**Funding acquisition:** Yan Sun.

**Investigation:** Yan Sun, Wenjie Hu, Xinqu Xia.

**Methodology:** Wenjie Hu.

**Project administration:** Yan Sun, Wenjie Hu.

**Software:** Wenjie Hu.

**Supervision:** Yan Sun, Xinqu Xia.

**Validation:** Yan Sun, Wenjie Hu, Xinqu Xia.

**Visualization:** Wenjie Hu.

**Writing – original draft:** Wenjie Hu.

**Writing – review & editing:** Wenjie Hu, Xinqu Xia.

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
