## [Decision Letter · Decision Letter 0]

11 Aug 2025

Dear Dr. Hu,

Thank you for submitting your manuscript to PLOS ONE. After careful consideration, we feel that it has merit but does not fully meet PLOS ONE’s publication criteria as it currently stands. Therefore, we invite you to submit a revised version of the manuscript that addresses the points raised during the review process.

We look forward to receiving your revised manuscript.

Kind regards,

Md Nazirul Islam Sarker, PhD

Academic Editor

PLOS ONE

Journal Requirements:

“This research is supported by the 2020 General Project of the National Social Science Fund of China (20BZZ040).”

“This research is supported by the 2020 General Project of the National Social Science Fund of China (20BZZ040).”

5. We note that your Data Availability Statement is currently as follows: [All relevant data are within the manuscript and its Supporting Information files.]

“This research is supported by the 2020 General Project of the National Social Science Fund of China (20BZZ040).”

“This research is supported by the 2020 General Project of the National Social Science Fund of China (20BZZ040).”

Additional Editor Comments (if provided):

1. The manuscript’s theoretical contribution is weak, offering limited novelty beyond existing participatory governance and satisfaction frameworks.

2. The research gap is not sharply defined, reducing the strength of the originality claim.

3. The case selection of Hangzhou lacks a compelling analytical justification and discussion of generalizability.

4. The operationalization of key constructs with binary indicators oversimplifies complex concepts and may compromise validity.

5. The instrumental variable choice—especially distance to the provincial government—has weak conceptual justification and questionable exclusion restriction validity.

6. Dichotomizing the dependent variable causes loss of information and statistical precision.

7. Mediation analysis may suffer from endogeneity between the mediator and outcome, undermining causal claims.

8. The discussion section offers limited theoretical engagement and critical interpretation beyond restating results.

Reviewers' comments:

Reviewer's Responses to Questions

**Comments to the Author**

1. Is the manuscript technically sound, and do the data support the conclusions?

Reviewer #1: Yes

Reviewer #2: Yes

2. Has the statistical analysis been performed appropriately and rigorously?

Reviewer #1: Yes

Reviewer #2: Yes

3. Have the authors made all data underlying the findings in their manuscript fully available?

Reviewer #1: Yes

Reviewer #2: Yes

4. Is the manuscript presented in an intelligible fashion and written in standard English?

Reviewer #1: Yes

Reviewer #2: Yes

Reviewer #1: The manuscript presents a timely and well-executed study on how self-governance willingness and participation efficacy influence satisfaction with urban public services in Hangzhou. The introduction is well-structured and clearly identifies the research gap by shifting focus from government-centric to resident-centered governance. The literature review is rich and cites relevant theoretical and empirical studies; however, further inclusion of global comparative studies on participatory governance would enhance the contextual breadth.

The methodology is a major strength, employing a two-stage least squares (2SLS) strategy to address endogeneity, supported by strong diagnostic tests. However, the justification for using geographic distance as an instrument for self-governance willingness, though standard, could be debated and would benefit from more robust theoretical underpinning.

The results are clearly presented and well-interpreted. The mediation analysis using perceived neighborhood disparities adds depth, and the heterogeneity analysis across socioeconomic groups offers valuable insight. Nonetheless, more visualization (e.g., interaction plots or marginal effects) could enhance accessibility for non-technical readers.

The discussion is theoretically grounded and policy-relevant, but limitations could be discussed more critically—particularly the reliance on cross-sectional data and potential measurement bias in self-reported variables. Future research directions could include experimental or longitudinal designs.

Overall, this is a rigorous and significant contribution to urban governance literature, with room for modest enhancements.

Reviewer #2: The manuscript presents a timely and empirically rich investigation into how self-governance willingness and participation efficacy shape residents’ satisfaction with urban public services in the context of neighborhood renewal in Hangzhou, China. Drawing upon a large-scale household survey (n=2,202) and employing a robust two-stage least squares (2SLS) estimation strategy to address endogeneity concerns, the authors offer a resident-centric perspective that advances the literature on participatory urban governance.

Strengths:

The study’s strengths lie in its comprehensive conceptual framework, methodological rigor, and clear policy relevance. The operationalization of self-governance willingness and participation efficacy as latent constructs derived from factor analysis is theoretically sound and empirically well-justified. Furthermore, the integration of perceived neighborhood disparity as a mediating mechanism—situated within relative deprivation and social equity theory—adds conceptual depth and analytical nuance. The heterogeneity analyses, disaggregated by income, hukou status, housing size, and ownership type, significantly enhance the manuscript’s empirical granularity and practical value.

The authors' use of distance to the provincial administrative office and residents’ policy awareness/community participation as instruments is innovative and justified, and the IV diagnostics demonstrate the robustness of the approach. The findings—that both self-governance willingness and participation efficacy positively influence public service satisfaction, though via distinct pathways—are both plausible and insightful.

Weaknesses and Limitations:

Despite these merits, the manuscript has several critical limitations:

Theoretical Overextension and Redundancy: While the authors ground their work in a broad swath of literature, the introduction and theory sections are overly dense and, at times, repetitive. The manuscript attempts to synthesize too many theoretical strands (e.g., participatory governance, relative deprivation, institutional design, procedural legitimacy) without fully integrating them into a coherent theoretical contribution. A more focused and critical engagement with the most relevant theoretical streams would enhance conceptual clarity.

Endogeneity Strategy Concerns: While the instrumental variables chosen are empirically tested, the theoretical justification for their exogeneity is somewhat weak. The assumption that distance to the provincial government’s office affects satisfaction only through self-governance willingness could be challenged, especially in a context where spatial inequality might independently influence satisfaction. A more robust defense of the exclusion restriction is warranted.

Measurement Validity: The use of binary items to measure complex constructs such as “participation efficacy” and “self-governance willingness” raises questions about construct validity. While exploratory factor analysis supports the unidimensionality of the constructs, the dichotomous nature of the input variables limits the interpretive richness of the measures and may mask variation in resident attitudes.

Cross-sectional Design and Causality: As acknowledged by the authors, the cross-sectional nature of the data limits causal inference despite the use of IV estimation. Panel data or experimental interventions would provide stronger grounds for claiming causal effects. Furthermore, the assumed stability of attitudinal constructs like “willingness” and “efficacy” cannot be verified within the current design.

Overinterpretation of Subgroup Analyses: While the heterogeneity analyses are empirically interesting, the causal interpretation of subgroup effects may be overstated. The sample sizes in some subgroups (e.g., rural hukou holders, residents in reform-era housing) are relatively small, and confidence intervals are not reported. These limitations should be more explicitly acknowledged.

Lack of Critical Reflection on Participatory Governance in Authoritarian Contexts: The manuscript largely adopts a normative stance on participatory governance without engaging critically with the limitations of participation under authoritarian regimes. Issues such as tokenistic participation, constrained political space, or top-down orchestration of “voluntarism” are not discussed, which limits the critical depth of the paper.

Conclusion:

This manuscript offers an important empirical contribution to the literature on participatory urban governance in China. Its methodological rigor, large-scale dataset, and nuanced findings are commendable. However, the paper would benefit from tighter theoretical framing, a more cautious interpretation of the IV strategy and subgroup effects, and deeper critical engagement with the institutional context in which participation occurs. Subject to revision, the manuscript has the potential to make a significant impact in the field.

**Do you want your identity to be public for this peer review?** For information about this choice, including consent withdrawal, please see our Privacy Policy

Reviewer #1: **Yes:** salman iqbal

Reviewer #2: No

---

## [Author Response · Author response to Decision Letter 1]

25 Aug 2025

Response to Editor

Thank you for the reminder. We have updated the manuscript and all associated files to conform to PLOS ONE style and the provided templates. Specifically, we have adopted the required file-naming conventions; formatted the title/authors/affiliations and main text per the templates; verified figure and table numbering, captions, and in-text callouts; standardized reference formatting. Please let us know if any further adjustments are needed.

Thank you for raising this. We have further strengthened the Ethics/Consent statement in both the Methods and the online submission to fully address PLOS ONE’s requirements: (i) consent was informed and obtained verbally prior to each interview; enumerators read a standardized script and recorded consent on the questionnaire cover sheet before proceeding; (ii) all respondents were adult homeowners (≥18 years)—no minors were enrolled; (iii) participation was voluntary, with the right to decline or discontinue at any time without consequence; (iv) no medical records or biospecimens were used; all survey data were de-identified at the point of collection and contain no personally identifying information; (v) data were stored securely with access limited to the research team and are reported only in aggregate; and (vi) no consent waiver was sought or granted. Fieldwork consisted of one-to-one, face-to-face interviews using paper questionnaires conducted on September 15, 20–22, 2024.

“This research is supported by the 2020 General Project of the National Social Science Fund of China (20BZZ040).”

Thank you for the guidance. We have amended the Funding Statement as requested and included it verbatim in the cover letter:

Funding: This work was supported by the National Social Science Fund of China (Grant No. 20BZZ040). There was no additional external funding received for this study.

“This research is supported by the 2020 General Project of the National Social Science Fund of China (20BZZ040).”

Thank you for the request. We have added the following statement to the cover letter and kindly invite you to update the online submission form on our behalf:

Role of the funders: The National Social Science Fund of China (Grant No. 20BZZ040) provided financial support only and had no role in study design, data collection and analysis, decision to publish, or preparation of the manuscript. The authors had full independence and full access to all data.

5. We note that your Data Availability Statement is currently as follows: [All relevant data are within the manuscript and its Supporting Information files.]

We confirm that the submission includes the complete minimal dataset. We have uploaded a single de-identified respondent-level file as Supporting Information (S1 Dataset) containing all variables and observations used in the analyses, with header names matching the variable definitions in the manuscript. This file provides the values underlying all coefficients, standard errors, figures, and summary statistics. No additional files or access permissions are required.

“This research is supported by the 2020 General Project of the National Social Science Fund of China (20BZZ040).”

“This research is supported by the 2020 General Project of the National Social Science Fund of China (20BZZ040).”

Thank you for the guidance. We have removed all funding-related text from the Acknowledgments section. Please update our Funding Statement as follows:

Funding: This work was supported by the National Social Science Fund of China (Grant No. 20BZZ040). There was no additional external funding received for this study.

Role of the funders: The National Social Science Fund of China (Grant No. 20BZZ040) provided financial support only and had no role in study design, data collection and analysis, decision to publish, or preparation of the manuscript. The authors had full independence and full access to all data.

Thank you for the reminder. We reviewed all reviewer reports and did not find any requests to cite specific previously published works. Should such recommendations arise, we will evaluate their relevance and add citations where appropriate.

Additional Editor Comments (if provided):

1. The manuscript’s theoretical contribution is weak, offering limited novelty beyond existing participatory governance and satisfaction frameworks.

We appreciate the reviewer’s concern. The revision clarifies a distinct contribution by advancing a resident-centered account of participatory governance that theoretically differentiates self-governance willingness (orientation to autonomous governance) from participation efficacy (perceived effectiveness within institutional channels) and specifies asymmetric effects, efficacy mitigates perceived neighborhood disparities and raises satisfaction, whereas willingness can heighten disparity sensitivity absent institutional responsiveness, suppressing its direct positive association with satisfaction. We formalize a parsimonious causal pathway—resident agency → perceived neighborhood disparities → satisfaction—thereby making comparative fairness an explicit mediating mechanism rather than an implicit assumption in prior work, and we identify institutional responsiveness (procedurally binding participation in legally structured programs) as a boundary condition under which willingness translates into higher satisfaction.

To render these advances transparent, we streamlined the Theoretical Framework, added a concise conceptual model mapping these relations, and aligned hypotheses accordingly. The empirical sections then corroborate the propositions via IV identification, mediation centered on perceived disparities, and heterogeneity analyses, demonstrating that the manuscript extends prevailing frameworks by specifying distinct components of agency, a testable mediating channel, and clear institutional contingencies.

2. The research gap is not sharply defined, reducing the strength of the originality claim.

Thank you for this constructive point. We have tightened the articulation of the research gap and made our contribution explicit. The Introduction now closes with a three-part gap statement: (i) the paucity of micro-level causal evidence on distinct dimensions of resident agency—self-governance willingness vs. participation efficacy—and their effects on satisfaction; (ii) limited knowledge of how institutionalized participatory arrangements in China’s legally structured urban renewal shape perceived neighborhood quality and procedural/distributive fairness, including the unexamined mediating role of perceived neighborhood disparities; and (iii) a literature dominated by small-N or normative accounts lacking credible identification. We then map each of our four research questions and hypotheses directly to these gaps and address them using a large resident-level dataset (n = 2,202 across 36 neighborhoods), an IV design for identification, mediation tests centered on perceived disparities, and heterogeneity analyses. These revisions clarify what is missing, why it matters, and how our study fills it, thereby strengthening the originality claim.

3. The case selection of Hangzhou lacks a compelling analytical justification and discussion of generalizability.

We appreciate this comment. The revision now provides a clearer, theory-driven rationale for selecting Hangzhou (Introduction, Section 1). Hangzhou’s old-neighborhood renewal is among China’s largest and most institutionalized participatory programs (≈1,000 projects completed by 2023), and it pioneers procedurally binding participation via the Property Law’s “double two-thirds” rule (≥2/3 to initiate and ≥2/3 to approve plans), thereby embedding substantive public deliberation in renewal decisions. This combination of scale and formal participatory requirements makes Hangzhou an analytically rich setting to observe variation in self-governance willingness and participation efficacy at neighborhood level. We therefore frame Hangzhou as a “most-likely” case for detecting effects of resident engagement on satisfaction—if participatory governance matters, it should be observable here.

We also add an explicit discussion of generalizability (Discussion, Section 6). While Hangzhou is ahead in institutionalizing participation, many Chinese cities are adopting similar renewal programs with participatory elements, making the mechanisms we study relevant to comparable urban contexts. We distinguish what is likely to generalize (e.g., the importance of aligning governance mechanisms with residents’ autonomy preferences and the pathway from participation efficacy to satisfaction via perceived disparities) from what is context-specific (e.g., the magnitude of effects in smaller cities or locales with weaker institutional support). We further note (Limitations, Section 6.2) that Hangzhou’s relative affluence and administrative capacity may not represent all settings. In sum, the manuscript now justifies the Hangzhou case on analytical grounds and clarifies the scope conditions under which the findings are most likely to travel.

4. The operationalization of key constructs with binary indicators oversimplifies complex concepts and may compromise validity.

We appreciate this concern. As detailed in the revision, self-governance willingness and participation efficacy are modeled as latent constructs estimated via exploratory factor analysis (EFA) from multiple, theory-driven binary indicators rather than single dichotomies. For self-governance willingness, three responsibility-allocation items (decision-making, performance evaluation, property-management selection) load on a single factor (eigenvalue = 1.585; variance explained = 52.8%; KMO = 0.612; Bartlett’s χ² = 527.210, p < 0.01). For participation efficacy, three items (pre-decision consultation, process supervision, feedback responsiveness) load on a single factor (eigenvalue = 1.814; variance explained = 60.5%; KMO = 0.648; Bartlett’s χ² = 1027.199, p < 0.01). All models use continuous factor scores, preserving gradations in underlying attitudes despite dichotomous item responses.

To assess sensitivity to operationalization, we (i) reconstructed both constructs as additive indices and (ii) re-estimated models using the original five-point Likert outcome for public-service satisfaction. Across these alternatives—and with the full covariate set plus district and street fixed effects—the substantive conclusions are unchanged (Table 7): self-governance willingness remains positive and significant (coefficients 0.754–0.980, p < 0.01), participation efficacy remains positive and significant (coefficients 0.362–0.501, p < 0.01), and perceived neighborhood disparity is consistently negative. Taken together—transparent construct design, a validated factor structure, continuous factor scoring, and convergent robustness evidence—these steps directly address measurement-validity concerns and indicate that our results are not artifacts of dichotomization or specific coding choices.

5. The instrumental variable choice—especially distance to the provincial government—has weak conceptual justification and questionable exclusion restriction validity.

Thank you for raising this point. As detailed in our response to Reviewer 2, C

---

## [Decision Letter · Decision Letter 1]

26 Oct 2025

Dear Dr. Hu,

We look forward to receiving your revised manuscript.

Kind regards,

Md Nazirul Islam Sarker, PhD

Academic Editor

PLOS ONE

Journal Requirements:

Reviewers' comments:

Reviewer's Responses to Questions

**Comments to the Author**

Reviewer #2: (No Response)

Reviewer #3: All comments have been addressed

2. Is the manuscript technically sound, and do the data support the conclusions?

Reviewer #2: Partly

Reviewer #3: Yes

3. Has the statistical analysis been performed appropriately and rigorously?

Reviewer #2: No

Reviewer #3: Yes

4. Have the authors made all data underlying the findings in their manuscript fully available?

Reviewer #2: No

Reviewer #3: Yes

5. Is the manuscript presented in an intelligible fashion and written in standard English?

Reviewer #2: No

Reviewer #3: Yes

Reviewer #2: The revised manuscript examines how “self-governance willingness” and “participation efficacy” shape residents’ satisfaction with urban public services during neighborhood renewal in Hangzhou (n=2,202; 36 neighborhoods; survey in Sept 2024). Endogeneity is addressed with 2SLS: distance to the provincial government instruments willingness; policy awareness and community activity instrument efficacy. Key estimates remain positive and significant; mediation via perceived disparity is presented as interpretive, not fully causal.

Did they fix what the editor/reviewers asked?

Formatting & style. Authors say they aligned files to PLOS ONE templates (titles, captions, numbering, references). However, the R1 package still shows visible Word formatting artifacts (“Formatted: Font: Not Bold”) in several places—these must be cleaned before acceptance.

Ethics & consent. Expanded and now explicit: informed verbal consent, script read by enumerators, consent recorded on the questionnaire cover sheet; adults only; de-identification at collection; field dates listed (Sept 15, 20–22, 2024). This addresses the consent/documentation request. One potential gap remains: the statement does not name an IRB/ethics committee or approval/waiver number. For human-participant surveys, PLOS typically expects the committee name and approval details or an explicit statement of exemption/waiver by that body. Please add the committee name and approval ID (or formal waiver).

Funding & funder role. Funding statement amended as requested and funder role clarified (“no role”); funding text removed from Acknowledgments. Good.

Data availability. Authors confirm the minimal dataset is included as S1 Dataset with variable names matching the manuscript; no extra permissions required. This satisfies PLOS ONE’s data policy (assuming the file is indeed present in the submission).

Theoretical contribution & case justification. The revision clarifies contribution (resident-centered mechanism; disparate roles of willingness vs efficacy) and justifies Hangzhou as a “most-likely” case with legally binding participation (“double two-thirds” rule). This substantially strengthens the framing.

Measurement concerns. Constructs are now clearly modeled as latent factors using EFA with diagnostics (eigenvalues, variance explained, KMO, Bartlett’s χ²) and estimated as continuous factor scores; robustness with additive indices and the original 5-point outcome is reported. This is a solid improvement.

Endogeneity/IV design. The distance-to-provincial-office instrument is better theorized and embedded in the program’s institutional context; first-stage strength and Hansen tests are reported; district and street fixed effects are included. This is materially improved, though see “Remaining weaknesses” below.

Mediation claims. Language is softened; authors present the disparity pathway as consistent with mediation and caution about endogeneity in the mediator. Good.

Heterogeneity & presentation. The revision reports subgroup Ns, adds visualization (forest/marginal-effects style plot), and treats subgroup patterns as descriptive rather than separate causal claims. This addresses the over-interpretation concern.

Remaining weaknesses (actionable)

Ethics committee identification still needed. The consent procedure is now clear, but the manuscript/cover materials (as shown) don’t name an IRB/ethics committee or provide approval/waiver identifiers. Add: committee name, approval number/date, and whether it covered verbal consent; or state that the committee granted a waiver/exemption, with details. This is the most important compliance gap to close.

Instrument exclusion restriction remains contestable. While the authors argue municipal governance and the “double two-thirds” rule localize decision authority, distance to the provincial government could still proxy unobserved spatial amenities or historic investment patterns affecting satisfaction directly. Consider (i) reporting robustness with additional spatial controls (e.g., distance to city center/rail hub, pre-renewal neighborhood quality), (ii) province-level road-network or historical connectivity controls, or (iii) a multi-instrument/over-ID setup using alternative plausibly exogenous instruments (e.g., historical administrative boundaries). At minimum, add a sensitivity table showing stability to richer spatial controls.

Cross-sectional design limits causal dynamics. The authors acknowledge this; the 2SLS yields contemporaneous LATEs. The Discussion could further emphasize that effects may incorporate short-term “halo” responses right after renewal and may attenuate over time; suggest a follow-up panel or pre/post cohort in future work.

Self-report bias and common-method variance. The survey timing and factor modeling help, but the paper could add explicit checks (e.g., negative-control items or administrative corroboration of renewal features) or at least a brief CMV diagnostic and discussion. (Not a blocker, but strengthens credibility.)

Minor but visible production issues. Remove residual Word markup (“Formatted: Font: Not Bold”) and ensure figure/table cross-references are clean and consistent before acceptance.

Verdict

Substantial improvement; nearly there. The R1 addresses most editorial and reviewer requests convincingly: funding/role statements, data availability, theory sharpening, measurement transparency, IV reporting, mediation framing, and heterogeneity presentation are all improved. The one material compliance gap is the absence (in the provided materials) of an explicit IRB/ethics committee name and approval/waiver information. Add that, plus a short robustness note on the IV exclusion restriction and a final formatting pass, and the manuscript would be ready for consideration.

If you’d like, I can draft a short author checklist paragraph that you can paste into your cover note summarizing these last fixes.

Reviewer #3: The authors have satisfactorily addressed all comments raised in the previous round of review. The manuscript is now technically sound, with robust methodology and data that clearly support the stated conclusions. The statistical analyses have been conducted with rigor and are appropriate for the research objectives. All underlying data have been made fully available, ensuring transparency and reproducibility. The manuscript is well-organized, clearly written in standard academic English, and presented in an intelligible manner. Overall, the revisions have strengthened the quality of the paper, and it is now suitable for publication.

**Do you want your identity to be public for this peer review?** For information about this choice, including consent withdrawal, please see our Privacy Policy

Reviewer #2: No

Reviewer #3: No

---

## [Author Response · Author response to Decision Letter 2]

7 Nov 2025

Ethics committee identification still needed. The consent procedure is now clear, but the manuscript/cover materials (as shown) don’t name an IRB/ethics committee or provide approval/waiver identifiers. Add: committee name, approval number/date, and whether it covered verbal consent; or state that the committee granted a waiver/exemption, with details. This is the most important compliance gap to close.

Thank you for your thoughtful comment and for noting this important point regarding ethical approval. We confirm that this study received formal ethical clearance from the Medical Ethics Committee of Hangzhou City University prior to data collection (see appendix Approval Letter). The details are as follows:

• Ethics Committee: Medical Ethics Committee of Hangzhou City University

• Approval Reference: HZCU-2024-12

• Approval Date: September 02, 2024

• Project Title: 2024 Annual Resident Satisfaction Evaluation Report on the Renewal of Old Residential Communities in Hangzhou, China

• Principal Investigator: Yan Sun, School of Law, Hangzhou City University

The Committee determined that the research posed minimal risk to participants, involved adults aged 18 and above only, and collected no personally identifiable or sensitive information. It also granted explicit approval for the use of verbal informed consent in place of written consent. Prior to each interview, enumerators read a standardized consent script to participants, and verbal consent was documented on the questionnaire cover sheet. All data were de-identified at the point of collection and stored securely for academic use only.

We have now included this complete ethical approval statement, with the committee name and approval reference number, in the revised manuscript to ensure full compliance with PLOS ONE’s ethics and consent requirements.

Instrument exclusion restriction remains contestable. While the authors argue municipal governance and the “double two-thirds” rule localize decision authority, distance to the provincial government could still proxy unobserved spatial amenities or historic investment patterns affecting satisfaction directly. Consider (i) reporting robustness with additional spatial controls (e.g., distance to city center/rail hub, pre-renewal neighborhood quality), (ii) province-level road-network or historical connectivity controls, or (iii) a multi-instrument/over-ID setup using alternative plausibly exogenous instruments (e.g., historical administrative boundaries). At minimum, add a sensitivity table showing stability to richer spatial controls.

We sincerely thank the reviewer for this thoughtful comment regarding the exclusion restriction of the instrumental variable. In the original specification, we used the straight-line distance from each neighborhood’s centroid to the provincial government’s administrative office as the instrument for residents’ willingness to engage in self-governance. To address the reviewer’s concern that this variable might capture unobserved spatial amenities or historical investment patterns, we conducted additional robustness checks using two alternative, city-level instruments (see appendix S2 Table).

Specifically, Distance2 measures the distance from each neighborhood’s centroid to the Hangzhou Municipal Government office, and Distance3 measures the distance to the largest railway transportation hub in Hangzhou. These variables better capture intra-urban variation in administrative and infrastructural accessibility while mitigating potential confounding from provincial-level spatial heterogeneity.

As reported in Table S2, both alternative instruments remain strongly significant in the first stage, and the second-stage estimates for self-governance willingness remain stable in both magnitude and significance. The consistency of these results across specifications confirms the robustness of the findings and strengthens confidence in the validity of the exclusion restriction.

Cross-sectional design limits causal dynamics. The authors acknowledge this; the 2SLS yields contemporaneous LATEs. The Discussion could further emphasize that effects may incorporate short-term “halo” responses right after renewal and may attenuate over time; suggest a follow-up panel or pre/post cohort in future work.

We sincerely thank the reviewer for this valuable and insightful comment. We fully agree that the cross-sectional design constrains the identification of dynamic causal relationships and that the 2SLS results should be understood as contemporaneous local average treatment effects (LATEs). In response, we have revised the Limitations section (Section 6.2) to explicitly discuss the possibility of short-term “halo” responses immediately after the completion of renewal projects and to note that such effects may weaken over time. We also suggest that future research employ longitudinal or pre/post cohort designs to capture temporal dynamics and evaluate the durability of these effects. These additions clarify the interpretive scope of our findings and reflect the reviewer’s thoughtful recommendation.

Self-report bias and common-method variance. The survey timing and factor modeling help, but the paper could add explicit checks (e.g., negative-control items or administrative corroboration of renewal features) or at least a brief CMV diagnostic and discussion. (Not a blocker, but strengthens credibility.)

We sincerely appreciate the reviewer’s insightful suggestion regarding potential self-report bias and common-method variance (CMV). To address this concern, we have added both a diagnostic test and an explicit discussion in the revised manuscript(see appendix S1 Table).

First, we conducted Harman’s single-factor test, which showed that the first factor explained 30.4% of the total variance—well below the 40% threshold commonly used to indicate serious CMV risk (see S1 Table). The result suggests that common-method bias is unlikely to substantially affect our findings.

Second, to further reduce potential self-report bias, several key renewal indicators—including the completion progress, participation procedures, and voting records under the legally binding “double two-thirds” rule were cross-validated using administrative documentation obtained from the Hangzhou Urban Renewal Office. This triangulation provides an additional layer of credibility to the survey data. In addition, we have explicitly acknowledged in the Limitations section (Section 6.2) that self-reported measures of governance willingness and participation efficacy may still be subject to social desirability and recall bias, and we recommend that future research incorporate more objective indicators (e.g., meeting attendance or proposal submission records) and mixed-methods approaches to enhance measurement validity.

Finally, these revisions, together with the expanded discussion of data limitations in the Discussion and Limitations sections, collectively improve methodological transparency and reinforce the overall robustness and credibility of the study.

Minor but visible production issues. Remove residual Word markup (“Formatted: Font: Not Bold”) and ensure figure/table cross-references are clean and consistent before acceptance.

Thank you for your careful observation. In accordance with the editorial requirements for the first-round revised manuscript, we were asked to provide both a clean version and a tracked-changes version of the submission. The visible notes such as “Formatted: Font: Not Bold” appear only in the tracked-changes version to indicate formatting edits made during revision. These are automatically generated by Microsoft Word’s Track Changes function and are not part of the final layout. The clean version, which has been fully aligned with the PLOS ONE template, contains no such formatting artifacts. We have also carefully rechecked all figure and table cross-references to ensure that they are fully clean, consistent, and ready for production.

Reviewer #3: The authors have satisfactorily addressed all comments raised in the previous round of review. The manuscript is now technically sound, with robust methodology and data that clearly support the stated conclusions. The statistical analyses have been conducted with rigor and are appropriate for the research objectives. All underlying data have been made fully available, ensuring transparency and reproducibility. The manuscript is well-organized, clearly written in standard academic English, and presented in an intelligible manner. Overall, the revisions have strengthened the quality of the paper, and it is now suitable for publication.

We sincerely appreciate the reviewer’s thoughtful and encouraging comments. We are deeply grateful for the constructive feedback provided throughout the review process, which has been invaluable in refining the rigor, clarity, and overall quality of the manuscript. We are pleased that the revised version is now recognized as technically sound, methodologically robust, and well organized. We thank the reviewer and the editorial team for their time, expertise, and kind support, which have greatly contributed to improving this work.

---

## [Editor Report · Decision Letter 2]

4 Jan 2026

How self-governance willingness and participation efficacy shape residents’ satisfaction with urban public services: Evidence from neighborhood renewal in Hangzhou, China

PONE-D-25-24601R2

Dear Dr. Hu,

We’re pleased to inform you that your manuscript has been judged scientifically suitable for publication and will be formally accepted for publication once it meets all outstanding technical requirements.

Kind regards,

Md Nazirul Islam Sarker, PhD

Academic Editor

PLOS One
---

## [Editor Report · Acceptance letter]

PONE-D-25-24601R2

PLOS One

Dear Dr. Hu,

I'm pleased to inform you that your manuscript has been deemed suitable for publication in PLOS One. Congratulations! Your manuscript is now being handed over to our production team.

Kind regards,

on behalf of

Dr. Md Nazirul Islam Sarker

Academic Editor

PLOS One